# THE HUMAN GENOMICS LONG-RANGE BENCHMARK: ADVANCING DNA LANGUAGE MODELS

## ABSTRACT

The advent of language models (LMs) in genomics necessitates benchmarks that can assess models' capabilities and limitations. In contrast to protein models, DNA LMs can be used to study non-coding regions of the genome and must account for unique challenges, especially interactions across long sequence lengths. However, existing benchmarks for DNA LMs are defined over short sequence datasets and can involve tasks that are not considered to be biologically meaningful. Here, we present the Human Genomics Long-Range Benchmark (LRB), which focuses on biologically meaningful tasks and supports long-range contexts. We complement our benchmark with fine-tuning recipes that meaningfully improve performance. We evaluate DNA LMs across nine compiled human genome tasks and observe that they achieve competitive performance relative to supervised baselines on several tasks (e.g., genome annotation), but there remains a significant gap in domains, such as variant effect and gene expression prediction. Additionally, we introduce a visualization tool to examine model performance split by genomic properties.

## 1 INTRODUCTION

Pre-training models on a large corpus of unlabeled data and subsequently fine-tuning to solve downstream tasks has demonstrated widespread success across domains, such as natural language processing (Achiam et al., 2023; Team et al., 2023) and computer vision (Oquab et al., 2023; Radford et al., 2021). This paradigm has also shown promise in biological applications, enabled by the wealth of unlabeled data coming from next-generation sequencing technologies. A prominent example are protein language models (LMs), which have been used to predict the effects of coding mutations on protein function (Lin et al., 2022), generate viable sequences conditioned on functional properties (Madani et al., 2023), and accurately predict structure from amino acid sequences (Lin et al., 2023). The development of these models has been made possible by benchmarks, such as CASP (Kryshtafovych et al., 2021), TAPE (Rao et al., 2019), PEER (Xu et al., 2022), and ProteinGym (Notin et al., 2023).

Genomics represents a new frontier for LMs in biology. The common pre-training tasks in language modeling (i.e., filling in missing tokens based on input context) inherently train LMs to model evolutionary forces, such as conservation and co-evolution, and the statistical patterns that these models learn can map to genomic motif identification, which is useful in accurate gene annotation. Indeed, significant progress has been made, with various LMs tailored to DNA sequences (Benegas et al., 2023a;b; Dalla-Torre et al., 2023; Ji et al., 2021; Nguyen et al., 2023; 2024; Schiff et al., 2024; Zhou et al., 2023). However, modeling genomic data presents unique challenges compared to proteomics. When modeling DNA, we have to account for non-coding regions and contend with interactions that can be orders of magnitude larger (Furlong and Levine, 2018). To guide the principled development of new DNA LMs, there is a need for robust benchmarks that accurately reflect these nuances. While several benchmarks have been proposed, these existing works contain important limitations. The vast majority of tasks proposed across existing benchmarks only consider short input contexts (less than 2k base pairs) (Dalla-Torre et al., 2023; Grešová et al., 2023; Marin et al., 2023; Zhou et al., 2023), disregarding long-range interactions that are highly impactful in genomics. Additionally, tasks in some benchmarks may be overly simplistic, failing to reflect real-world use cases, e.g., some benchmarks have used synthetic data to construct negative sets (Dalla-Torre et al., 2023).

To bridge these gaps, we propose the Human Genomics Long-Range Benchmark (LRB), a compilation of biologically meaningful tasks in human genomics. Our benchmark deliberately incorporates tasks hypothesized to span both short and long genomic contexts. Allowing users to select arbitrary sequence length inputs for any given dataset enables us for the first time to understand empirically the importance of long-range inputs for our proposed tasks. We also include available genomic annotations and provide a visualization tool that allows users to analyze results in more detail. We demonstrate the benefit of full model fine-tuning compared to previous approaches that keep backbone DNA LM weights frozen during downstream training. To summarize, we make the following contributions:

**1. Release the Genomics Long-Range Benchmark**, composed of biologically meaningful tasks that cover both short- and long-range genomic scales. Our benchmark is unique in that it allows users to download arbitrary length sequences for each task. We provide evaluation results for a selection of prominent DNA LMs in both zero-shot and fine-tuning settings along with comparisons against reference baselines. We find that on genomic annotation tasks DNA LMs perform competitively with existing supervised models, but on the long-range prediction tasks of gene expression and zero-shot mutation effect prediction there persists a large gap, especially when compared to alignment based LMs.

**2. Develop and analyze improved fine-tuning methods** that better reflect real-world usage, finding that full model weight fine-tuning significantly improves performance.

**3. Introduce an analysis and visualization tool** to examine models' performance across different genomic properties. This tool enables deeper analyses that reveal more nuanced evidence that DNA LMs lag behind a well-regarded and long-range supervised baseline, Enformer (Avsec et al., 2021a), in modeling long-range interactions.

## 2 BACKGROUND

### 2.1 LANGUAGE MODELING FOR DNA

Supervised machine learning methods have been successfully applied to genomics (Alipanahi et al., 2015; Avsec et al., 2021a; de Almeida et al., 2022; Zhou and Troyanskaya, 2015; Zhou et al., 2018b). However, these models depend on large amounts of labeled data and tend to be task-specific. LMs have recently

Table 1: Comparison to existing benchmarks.

| | Long range | Human centric | Biological relevance |
|---|---|---|---|
| NT (Dalla-Torre et al., 2023) | ✗ | ✗ | ✗ |
| GB (Grešová et al., 2023) | ✗ | ✗ | ✗ |
| GUE (Zhou et al., 2023) | ✗ | ✗ | ✔ |
| BEND (Marin et al., 2023) | ✗ | ✔ | ✔ |
| **Genomics LRB** *(Ours)* | ✔ | ✔ | ✔ |

gained traction in the genomics domain: the abundance of unlabeled sequences supports robust model pre-training and the widely-used pre-training objectives of next token prediction (NTP) or masked language modeling (MLM) directly lend themselves to models identifying genomic motifs and evolutionary patterns, e.g., conservation. Some notable recent works include DNABERT (Ji et al., 2021; Zhou et al., 2023; 2024), GPN (Benegas et al., 2023a;b), Nucleotide Transformer (NT) (Dalla-Torre et al., 2023), GENA-LM (Fishman et al., 2023), HyenaDNA (Nguyen et al., 2023), Evo (Nguyen et al., 2024), Evo2 (Brixi et al., 2025), and Caduceus (Schiff et al., 2024). A more thorough review of recent DNA LMs is deferred to Section A.2.

### 2.2 DNA LM EVALUATION

The goal of DNA LMs is to learn meaningful representations that can be used to improve performance on downstream tasks. Existing benchmarks, which include the Nucleotide Transformer tasks (NT; Dalla-Torre et al. (2023)), Genomic Benchmark (GB; Grešová et al. (2023)), Genome Understanding Evaluation (GUE; Zhou et al. (2023)), and Benchmark for DNA LMs (BEND; Marin et al. (2023)), have been crucial for establishing baseline model capabilities. However, these benchmarks contain several important shortcomings: they do not focus on long-range sequences, they can contain synthetic examples, and their evaluations do not take full advantage of pre-trained models. See Section A.3 for a more complete description of existing works.

## 3 THE GENOMICS LONG-RANGE BENCHMARK

Table 2: Overview of the tasks contained in the Genomics Long-Range Benchmark.

|  | Type | # Outputs | # Train / Test | % Pos. Label |
|---|---|---|---|---|
| *Variant Effect Prediction* | | | | |
| Causal eQTL | SNP Classification | 1 | 89k / 9k | 50.0 |
| Pathogenic OMIM | SNP Classification | 1 | - / 2.3M | 0.02 |
| Pathogenic ClinVar | SNP Classification | 1 | 39k / 1k | 56.1 |
| *Gene Expression Prediction* | | | | |
| Bulk RNA-seq | Seq-wise Regression | 218 | 23k / 1k | - |
| CAGE profile | Binned Regression | 50 / bin | 34k / 2k | - |
| *Regulatory Element Detection* | | | | |
| Promoter | Seq-wise Classification | 1 | 953k / 96k | 4.7 |
| Enhancer | Seq-wise Classification | 1 | 1.9M / 192k | 52.5 |
| *Chromatin Feature Identification* | | | | |
| Histone Mark Prediction | Seq-wise Classification | 20 | 2.2M / 227k | 7.0 |
| Chromatin Accessibility | Seq-wise Classification | 20 | 2.2M / 227k | 4.4 |

Below we describe the nine tasks that we compiled from various human genome data sources that comprise our proposed Genomics Long-Range Benchmark (LRB). Our suite consists of tasks that are hypothesized to require only short-range contexts as well as those thought to need longer sequences for accurate prediction. By enabling users to download data at arbitrary length scales (the first benchmark to support this feature), these hypotheses can be rigorously tested. Our tasks span various applications that are of interest to practitioners, namely variant effect prediction, gene expression prediction, regulatory element detection, and chromatin factor identification; see Table 2. Below, for each task, we provide details on the biological relevance that motivated its inclusion, a formal task definition, and rationale for hypothesized long-range dependencies (where applicable). We defer additional details, e.g., data source and processing, train / test splits, and metric definition, to Section B.

### 3.1 VARIANT EFFECT PREDICTION

#### 3.1.1 CAUSAL EQTL

**Biological Relevance** Predicting the effects of genetic variants, particularly expression quantitative trait loci (eQTLs), is essential for understanding the molecular basis of several diseases. eQTLs are genomic loci that are associated with variations in mRNA expression levels among individuals. By linking genetic variants to causal changes in mRNA expression, researchers can uncover how certain variants contribute to disease development (Consortium, 2020).

**Task Definition** The task is formulated as a binary classification problem to distinguish eQTLs from GTEx (Consortium, 2020) from a set of matched negatives identified in Avsec et al. (2021a). Inputs are sequences centered around candidate single nucleotide polymorphisms (SNPs) each assigned a causal probability by fine-mapping using the "Sum of Single Effects" (SuSiE) model (Wang et al., 2020). Following Avsec et al. (2021a), variants with causal probability greater than 0.9 are labeled as positive and variants with causal probability less than 0.01 are labeled as negative.

**Long-Range** The regulation of gene expression is modulated by distal, cis-regulatory elements, called enhancers, that can be more than several hundred thousand base pairs (bps) away from a target gene (Furlong and Levine, 2018). Variants that impact gene expression are often located at such distal elements, and thus, to predict such variants, models should have long context windows (Avsec et al., 2021a).

#### 3.1.2 PATHOGENIC OMIM

**Biological relevance** Predicting the effects of regulatory variants on pathogenicity is crucial for understanding disease mechanisms (Marwaha et al., 2022). Elements that regulate gene expression

Table 3: Benchmarking performance of DNA LMs and baselines on variant effect prediction tasks. Models were evaluated using both fine-tuning and zero-shot. Best sequence-based LM values are **bolded**. †Evo2 was only evaluated on 512bp sequences due to computational constraints.

| Model Name | Context (bps) | Causal eQTL (AUROC) | | Path. ClinVar (AUROC) | | Path. OMIM (AUPRC) |
|---|---|---|---|---|---|---|
| | | Fine-tune | Zero-shot | Fine-tune | Zero-shot | Zero-shot |
| *DNA LMs* | | | | | | |
| DNABERT-2 | 10k | 0.72 ± 0.008 | 0.50 ± 0.010 | 0.74 ± 0.013 | 0.50 ± 0.02 | 0.002 ± 0.0002 |
| DNABERT-S | 10k | **0.73** ± 0.008 | - | 0.73 ± 0.011 | - | - |
| NTv2 500M | 12k | 0.72 ± 0.003 | **0.51** ± 0.008 | **0.78** ± 0.009 | 0.68 ± 0.01 | 0.003 ± 0.0011 |
| HyenaDNA 160K | 160k | 0.71 ± 0.010 | 0.51 ± 0.013 | 0.56 ± 0.073 | 0.49 ± 0.02 | 0.002 ± 0.0002 |
| Evo2 7B | 512† | - | 0.50 ± 0.008 | - | **0.89** ± 0.01 | 0.054 ± 0.0100 |
| *Alignment-based LM* | | | | | | |
| GPN-MSA | 128 | 0.70 ± 0.008 | 0.55 ± 0.010 | 0.97 ± 0.01 | 0.97 ± 0.01 | 0.34 ± 0.03 |
| *Baseline* | | 0.76 ± 0.002 (Enformer) | 0.56 (CADD) | 0.65 ± 0.031 (Enformer) | 0.97 ± 0.01 (CADD) | 0.208 ± 0.02 (CADD) |

Table 4: Benchmarking performance of DNA LMs and baselines on gene expression, regulatory element, and chromatin features tasks. Models were evaluated in only a fine-tuned setting for this set of tasks. Best sequence-based LM values are **bolded** and in **green** if LM beats baseline.

| Model Name | Context (bps) | Bulk RNA ($R^2$) | CAGE ($R^2$) | Promoter (AUPRC) | Enhancer (AUROC) | Histone Marks (AUPRC) | DNA Accessibility (AUPRC) |
|---|---|---|---|---|---|---|---|
| *DNA LMs* | | Fine-tune | Fine-tune | Fine-tune | Fine-tune | Fine-tune | Fine-tune |
| DNABERT-2 | 10k | 0.51 ± 0.050 | - | 0.71 ± 0.112 | 0.81 ± 0.022 | 0.24 ± 0.091 | 0.15 ± 0.064 |
| DNABERT-S | 10k | 0.52 ± 0.060 | - | 0.75 ± 0.021 | **0.83** ± 0.005 | 0.33 ± 0.006 | 0.16 ± 0.039 |
| NTv2 500M | 12k | **0.60** ± 0.038 | **0.39** ± 0.011 | **0.79** ± 0.006 | 0.82 ± 0.002 | **0.38** ± 0.003 | **0.3** ± 0.007 |
| HyenaDNA 160K | 160k | 0.46 ± 0.006 | 0.19 ± 0.032 | 0.67 ± 0.009 | 0.74 ± 0.009 | 0.25 ± 0.004 | 0.11 ± 0.002 |
| *Alignment-based LM* | | | | | | | |
| GPN-MSA | 128 | - | 0.09 ± 0.012 | 0.73 ± 0.015 | 0.79 ± 0.005 | - | - |
| *Baseline* | | 0.83 ± 0.005 (Enformer) | 0.49 ± 0.000 (Enformer) | 0.86 ± 0.006 (Enformer) | 0.92 ± 0.002 (Enformer) | 0.35 (DeepSea) | 0.44 (DeepSea) |

are often located in non-coding regions, and variants in these areas can disrupt normal cellular function, leading to disease. Accurate predictions can identify biomarkers and therapeutic targets, enhancing personalized medicine and genetic risk assessment.

**Task Definition** The task is formulated as a binary classification problem where inputs are DNA sequences centered around a SNP and outputs are binary labels. The dataset was constructed following Benegas et al. (2023a), where the negative class corresponds to a common (mean allele frequency > 5%) SNP in gnomAD (Chen et al., 2022) and the positive class corresponds to a pathogenic SNP, defined as a SNP in a regulatory region having an implication in a Mendelian disorder in the Online Mendelian Inheritance in Man database (Smedley et al., 2016).

**Long-Range** Regulatory elements like enhancers and silencers can exist far from the genes they regulate (Furlong and Levine, 2018). Variants in these regulatory elements can lead to aberrant gene expression patterns and ultimately disease, but identifying such regulatory variants is difficult since regulatory elements can modulate the expression of proximal or distal genes. Models that can capture interactions between possibly distal regulatory elements and their target genes while still being able to capture the proximal interactions are essential to identifying non-coding pathogenic variants.

### 3.1.3 PATHOGENIC CLINVAR

**Biological Relevance** A coding variant refers to a genetic alteration that occurs within the protein-coding regions of the genome, also known as exons. Such alterations can impact protein structure, function, stability, and interactions with other molecules, ultimately influencing cellular processes and potentially contributing to the development of genetic diseases (Lek et al., 2016). Predicting variant

pathogenicity is crucial for guiding research into disease mechanisms and personalized treatment strategies, enhancing our ability to understand and manage genetic disorders effectively.

**Task Definition** This task is formulated as a binary classification with inputs centered around SNPs. The dataset was constructed following Benegas et al. (2023a), where the negative class corresponds to a common (minor allele frequency $> 5\%$) SNP in gnomAD (Chen et al., 2022) and the positive class to pathogenic SNPs identified in ClinVar (Landrum et al., 2020).

## 3.2 GENE EXPRESSION PREDICTION

### 3.2.1 BULK RNA-SEQ

**Biological Relevance** Gene expression involves the process by which information encoded in a gene directs the synthesis of a functional gene product, typically a protein, through transcription and translation. Transcriptional regulation determines the amount of mRNA produced, which is then translated into proteins. Models that can predict RNA expression levels solely from sequence data are can advance our understanding of gene regulation, elucidate disease mechanisms, and identify functional sequence variants.

**Task Definition** This task is described as a multi-variable, sequence-wise regression task. Data was constructed following Zhou et al. (2018a) such that inputs are DNA sequences centered around the transcription start site (TSS) of each gene where the TSS was identified using a combination of annotations from GENCODE (Harrow et al., 2012) and CAGE data from FANTOM5 (Forrest et al., 2014). Outputs are RPKM normalized RNA expression counts for each gene obtained from Consortium (2020) that were $\log(1 + x)$ normalized and standardized. For each gene, there are 218 different counts corresponding to the RNA expression level in different tissue types.

**Long-Range** RNA gene expression is regulated by non-coding elements, such as enhancers and silencers, which can be located hundreds of kilo-bps away from the gene (Furlong and Levine, 2018), indicating the possible presence of long-range interactions in transcription regulation.

### 3.2.2 CAP ANALYSIS GENE EXPRESSION (CAGE) PROFILE

**Biological Relevance** CAGE provides accurate high-throughput measurements of RNA expression by mapping TSSs at a nucleotide-level resolution (Takahashi et al., 2012). This is vital for detailed mapping of TSSs, understanding gene regulation mechanisms, and obtaining quantitative expression data to study gene activity comprehensively.

**Task Definition** This task is described as a multi-variable, binned nucleotide-wise regression task. The data was constructed following the approach outlined in Basenji (Kelley, 2020). Inputs are DNA sequences and the outputs are $\log(1 + x)$ normalized CAGE expression counts from FANTOM5 (Forrest et al., 2014) given for each 128 bp bin of the input sequence. For each bin in a sequence, there are 50 different values corresponding to expression amounts across 50 human cell / tissue types.

**Long-Range** The production of RNA via transcription as measured by CAGE is regulated by non-coding elements that can be located hundreds of kilo-bps away from the gene, indicating the presence of long-range interactions in transcription regulation (Furlong and Levine, 2018).

## 3.3 CIS-REGULATORY ELEMENT DETECTION

**Biological Relevance** Cis-regulatory elements, such as promoters and enhancers, control the spatial and temporal expression of genes (Andersson and Sandelin, 2020). These elements are essential for understanding gene regulation mechanisms and how genetic variations can lead to differences in gene expression.

**Task Definition** This task is described as a binary classification problem. Data from Search Candidate Regulatory Elements by ENCODE (SCREEN (The ENCODE Project Consortium, 2020)) was processed according to our approach outlined in Section B.3. Inputs are sequences sampled from across the entire human genome and outputs are binary values, with positive labels assigned to sequences if their center 200 bps overlap by at least 50% with an annotated enhancer or promoter. This task is composed of two sub-tasks: (1) predicting the presence of promoters and (2) predicting the presence of enhancers.

### 3.4 CHROMATIN FEATURE IDENTIFICATION

**Biological Relevance** Chromatin features (histone marks and DNA accessibility) are crucial for understanding gene regulation, as these features indicate chromatin state and are essential for transcription activation (Zhou et al., 2018b).

**Task Definition** This task is a multi-label binary classification problem constructed following Zhou and Troyanskaya (2015), where sequences were sampled from the human genome as inputs and outputs correspond to binary labels for different chromatin profiles. The task contains two sub-tasks: one for predicting histone marks and another for predicting chromatin accessibility. For histone marks, each of the 20 binary values represents a different histone mark in a specific cell type. For DNA accessibility, each of the 20 binary values corresponds to a different tissue/cell type. A value is labeled as positive if the center 200 bps of the input sequence overlaps by at least 50% with a peak region measured by ChIP-seq (histone marks) or DNase-seq (DNA accessibility) obtained from ENCODE and the Roadmap Epigenomics consortium (Bernstein et al., 2010; The ENCODE Project Consortium, 2020).

### 3.5 IMPROVED EVALUATION WITH FULL FINE-TUNING

To evaluate DNA LMs we perform fine-tuning, i.e., we train a model in a supervised manner on a downstream task. Our fine-tuning strategy involves extracting embeddings from each model which are then input to a task-specific prediction head (see Section D for details). In previous benchmarks, authors fine-tuned models by freezing the embeddings (Marin et al., 2023). We perform a systematic study of fine-tuning strategies and discover that this strategy significantly hurts DNA LM performance. We therefore provide a recipe for full-parameter fine-tuning and show that it significantly improves performance across many tasks, enabling us to evaluate models more fairly than in previous works and setting new best-practices for DNA LMs (independent of our benchmark).

### 3.6 ADDITIONAL NOVEL FEATURES OF THE LRB

In addition to our careful curation of tasks and improved fine-tuning, we highlight two more novel aspects.

**Visualization Tool** We provide benchmark users with a visualization tool in the form of an interactive `jupyter` (Kluyver et al., 2016) notebook. To create this tool, we collected genomic annotation datasets from SCREEN, GENCODE, RepeatMasker (Harrow et al., 2012; Smit et al., 2015; The ENCODE Project Consortium, 2020) and aligned them to our benchmark task datasets; see Section B.5 for details and screenshots. Our tool enables a deeper level of analysis compared to what other benchmarks afford. For example, users can view models' performance in aggregate, by specific annotations, and also by distance to TSSs.

**Arbitrary Sequence Length** Our benchmark allows users to download arbitrary sequence lengths for any given tasks. This enables the probing of the effect of sequence length and lets users evaluate their LMs on the same context size on which they performed pre-training.

### 3.7 SELECTED BASELINES

To contextualize the performance of DNA LMs, we curate a set of task-specific expert methods that are comprised of well-regarded supervised models.

**Combined annotation dependent depletion** (CADD) (Schubach et al., 2024) is a SVM developed for detecting deleterious DNA variants. We use this method as an expert baseline for our zero-shot variant effect prediction tasks.

**GPN-MSA** Benegas et al. (2023a) present an alignment-based DNA LM for variant effect prediction based on the RoFormer (Su et al., 2021) architecture. In addition to the standard input DNA sequence, a Multiple Sequence Alignment (MSA), an alignment of similar sequences from multiple species, is used as an input. This alignment is computed from 89 vertebrates and is always unmasked, giving the model access to evolutionary information for a given input sequence. The auxiliary alignment information and strong performance on zero-shot prediction render GPN-MSA a useful comparison against sequence-based LMs.

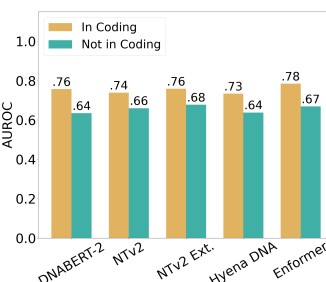 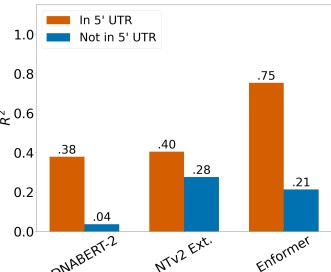 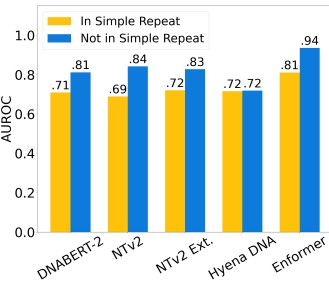

(a) Fine-tuned Causal eQTL prediction; by protein coding annotation.

(b) Bulk RNA prediction; by 5′ UTR annotation

(c) Enhancer Annotation; by simple repeat annotation.

Figure 1: Results split by genomic annotations.

**Enformer** (Avsec et al., 2021a) is composed of both convolutional and transformer layers and trained in a supervised manner on various biological tasks using a context length of up to 196k bps. We use Enformer as the expert method for fine-tuning versions of variant effect prediction, gene expression prediction, and regulatory element detection tasks.

**DeepSEA** (Zhou et al., 2018b) is a convolutional network trained to predict chromatin profile data, such as transcription factor binding, histone marks, and DNA accessibility. As our chromatin feature tasks are derived from DeepSEA, we use it as the expert method for these tasks.

## 4 RESULTS

### 4.1 EXPERIMENTAL SETUP

We evaluate several prominent DNA LMs on our benchmark: the Nucleotide Transformer v2 (NTv2) series (Dalla-Torre et al., 2023), a context length extended version of NTv2 (NTv2-Ext) (see Section C) DNABERT-1 (Ji et al., 2021), DNABERT-2 (Zhou et al., 2023), the HyenaDNA series (Nguyen et al., 2023), Evo2 (Brixi et al., 2025), and Caduceus (Schiff et al., 2024) representing a range of pre-training datasets and objectives, architectures, and model sizes. For fine-tuning, we use an MLP as the prediction head and train both the DNA LM and MLP weights (see Section D for full details).

For classification tasks with highly imbalanced labels (see Table 2), we use area under precision-recall curve (AUPRC) as opposed to receiver operator curve (AUROC).

**Fine-tuning** Models are trained using either mean-squared error loss for regression tasks or cross-entropy loss for classification tasks. For each task, we perform five-fold cross-validation (CV) using different random seeds, where we create different train / validation splits, select the best-performing model using early stopping on validation loss, and evaluate it on the held-out test set. We report the mean ± standard deviation performance across folds as final metrics.

Table 5: Difference in performance by fine-tuning strategy. Results correspond to fine-tuning NTv2 50M on a selection of the benchmark tasks.

|  | **Path. ClinVar** *(AUROC)* | **Bulk RNA** ($R^2$) | **Promoter** *(AUPRC)* |
|---|---|---|---|
| Head Only | 0.69 ±0.003 | 0.44 ±0.008 | 0.73 ±0.001 |
| Peft | 0.70 ±0.002 | 0.47 ±0.002 | 0.74 ±0.001 |
| Full FT | **0.75** ±0.001 | **0.54** ±0.001 | **0.76** ±0.003 |

**Zero-Shot Prediction** We also evaluate the zero-shot performance on our three variant effect prediction tasks to account for the fact that determining the pathogenicity or causality of variants is difficult, which often results in smaller datasets not suitable for fine-tuning. To establish error bars for our zero-shot tasks, we bootstrapped the data $N = 1000$ times and report the mean and standard deviation of the metrics over all iterations. Given the extreme class imbalance in the Pathogenic OMIM dataset, we only perform zero-shot evaluation for this task.

### 4.2 Main DNA LM Results

In Tables 3 and 4, we present the top performing DNA LMs (full results in Section E).

**Variant Effect Prediction** For zero-shot evaluation, we observe that DNA LMs are outperformed by the CADD and GPN-MSA baselines on all variant effect prediction tasks. Additionally, for zero-shot Causal eQTL, we find that all models struggle, with near-random performance. Predicting pathogenicity, is the clearest example where DNA LMs except for Evo2 fall short of CADD and GPN-MSA, which have nearly 2x better performance in ClinVar and about 100x in OMIM compared to non Evo2 models. Even then, Evo2 still lags behind these baselines on the configuration we were able to evaluate and is generally very computationally expensive to use. When fine-tuning, we find that DNA LM performance on both variant tasks greatly improves, matching or surpassing the strong Enformer baseline. Importantly, the alignment-based GPN-MSA model, despite using short context inputs (128 bps), outperforms CADD and all single-sequence DNA LMs, highlighting the importance of capturing conservation in predicting pathogenic variant effects. For DNA LMs to be useful for these tasks, they must also find a way to model and learn evolutionary pressures and conservation.

**Gene Expression Prediction** While NTv2 is the best performing DNA LM for Bulk RNA and CAGE tasks, the baseline Enformer outperforms LMs by a wide margin.

**Regulatory Element Detection** DNA LMs are able to accurately predict the presence of regulatory elements, especially considering the class-imbalance present in promoter detection, with NTv2 performing best among DNA LMs. However, there remains a gap to the Enformer model.

**Chromatin Feature Identification** For both histone mark and DNA accessibility, NTv2 is significantly better performance than the other DNA LMs, and even exceeds the supervised baseline on the former task.

### 4.3 Analyzing Results by Genomic Annotations

We developed an analysis and visualization tool to examine models performance across different genomic properties and annotations. Using our tool we are able to perform deeper analyses and extract insights about the performance of each model, which are inaccessible to users of existing benchmarks. We detail some examples in Figure 1.

**Causal eQTL Prediction (Fine-tune)** By stratifying SNPs into protein-coding and non-coding regions in Figure 1a, we find a potential failure mode for both DNA LMs and supervised models. Non-coding variants presumably entail regulatory and possibly longer-range interactions, and all models perform worse in these regions.

**Bulk RNA Expression Prediction** In Figure 1b, we see that the performance of DNA LMs and Enformer drops precipitously when focusing on non-5′ regions that likely entail longer-range interactions. However, we also observe that the context-extended NTv2 outperforms Enformer on this region, implying that the majority of the performance gap between DNA LMs and the Enformer baseline lies in modeling variants in the 5′ regions.

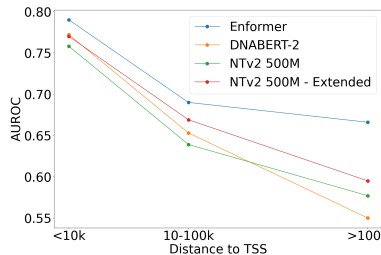

**Enhancer Detection** In Figure 1c, we observe that most models, including Enformer, suffer a performance hit when identifying enhancers within simple repeats, likely due to the difficulty of detecting enhancers in repetitive regions.

Figure 2: Fine-tuned Causal eQTL variant task; by distance to nearest TSS.

### 4.4 Effect of Fine-tuning Methodology

In Table 5, we demonstrate the importance of our proposed fine-tuning. For NTv2 (see additional results in Section E.4), we show how full fine-tuning, as opposed to freezing LM weights and only training a prediction head, a common practice in existing benchmarks such as BEND (Marin et al.,

2023), drastically improves model performance. Furthermore, we show the difference between full-finetuning and parameter-efficient finetuning such as $(IA)^3$ (Liu et al., 2022). We believe our methodology is more in line with how practitioners would use DNA LMs in real-world settings, but we also show that the choice of fine-tuning methodology can in itself result in significant differences between model performance.

## 4.5 IMPORTANCE OF CONTEXT LENGTH

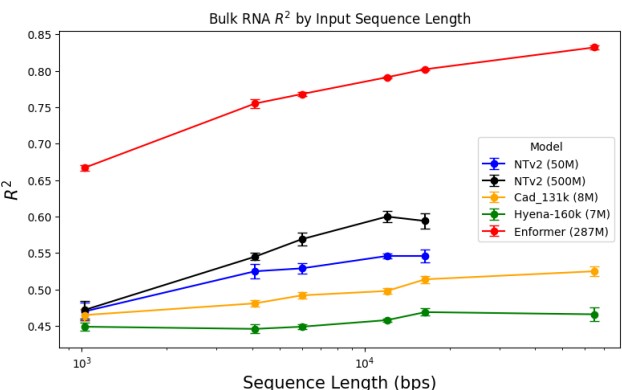

Figure 3: Performance ($R^2$) on the Bulk RNA prediction task for varying input sequence lengths. NTv2 models ran out of memory at 16k bps.

To verify our hypothesis that the long-range tasks in our benchmark benefit from larger context sizes we fine-tune the Enformer baseline (287M) and other DNA LM models (Hyena (7M), NTv2 (50M, 500M), and Caduceus (8M) for various input sequence lengths. In Figure 3 we find that on the Bulk RNA task, the input sequence length is a major component driving model performance, with the Enformer baseline having significantly worse performance at small (1k) input lengths ($R^2 = 0.67$) compared to longer (65k) input lengths ($R^2 = 0.83$). Additionally, while most DNA LMs saw an increase in performance with increasing context length, all of the assayed DNA LMs are considerably worse than the Enformer baseline on all context lengths, even the larger NTv2 500M model. Even at small context lengths, there is a considerable performance gap between DNA LMs and existing methods, and this gap seems to become more apparent as the context lengths grows, highlighting opportunities for further development of DNA LMs.

In addition, in Figure 2 we conduct some further analysis on the causal eQTL and group the variants by the distance to their closest annotated TSS. We observe an inverse relationship between this distance and the performance on those variants, and that the longer context models retain their performance at the furthest bin. This likely further indicates that this task captures 'long-range' interactions.

## 5 DISCUSSION AND CONCLUSION

In this work, we introduced the Human Genomics LRB. Our benchmark is the first to truly evaluate long-range capabilities. We provided initial results for several prominent DNA LMs, with more in-depth analysis than previous benchmarks explored. Our results demonstrate the importance of fully fine-tuning models. Additionally, we identify several domains where a large performance gap needs to be bridged before DNA LMs can be reliably used and some failure modes of DNA LMs. Namely, zero-shot DNA LM variant effect prediction is not yet mature enough to replace widely-used tools, such as CADD or alignment-based models like GPN-MSA. Similarly, for gene expression prediction, DNA LMs lag far behind supervised methods. These results demonstrate that future DNA LM efforts should focus on the more difficult tasks that entail long-range interactions, and we hope that our benchmark spurs such development.

**Future Work** One potential limitation of our work is the lack of hyperparameter search for fine-tuning; a more extensive search would better differentiate models. Additionally, very large models, such as Evo2, were only evaluated on zero-shot tasks due to high computational cost of running these models. In future iterations of our benchmark, we also plan to add more tissue-specific analyses, bp-level annotation tasks, and tasks covering multiple species.

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

## A    EXTENDED BACKGROUND

### A.1    TERMINOLOGY

The genome is a sequence of four nucleotides (*Adenine*, *Cytosine*, *Thymine*, and *Guanine*) organized into a double-stranded helical structure called *deoxyribonucleic acid* (DNA). This structure encodes the information required for the development, maintenance, and function of cells. Genetic information flows from DNA to *messenger ribonucleic acid* (mRNA) by a process called *transcription*, and mRNA is used as a blueprint to create *proteins* via a process called *translation*. Proteins are responsible for initiating and sustaining the cellular processes, while DNA encodes the information necessary for their production.

The genome is organized into functional elements, including *coding* and *non-coding* regions. Coding regions comprise genes responsible for protein synthesis, while non-coding regions can play vital regulatory roles. *Promoters*, a type of regulatory region, are situated close to genes and serve as sites for transcription initiation. *Enhancers*, another regulatory element located farther from genes, modulate gene expression by recruiting transcription factors, a type of protein that regulates transcription. Notably, a single gene can be regulated by multiple promoters and enhancers simultaneously.

DNA does not exist solely as a linear molecule but is instead tightly packaged around *histone* proteins, forming a sphere of wound DNA called *nucleosomes*. These nucleosomes further assemble into *chromatin*, which constitutes the 23 pairs of *chromosomes* in humans. Chromatin can exist in an open (*euchromatin*) or closed (*heterochromatin*) state, influencing the ability of the underlying DNA to be transcribed. Chemical modifications to histones play significant roles in chromatin remodeling acting as signals that recruit proteins to either condense the chromatin structure (making it less accessible) or relax it (making it more accessible), thereby influencing gene activity.

Mutations in the genome, including *single nucleotide polymorphisms* (SNPs), insertions, and deletions, can alter DNA sequences, potentially disrupting functional genomic elements or affecting the structure and function of proteins. Understanding the impact of these sequence variations on disease remains a central challenge in biology. Such mutations can lead to genetic disorders or contribute to the development of complex diseases.

### A.2    RECENT DNA LMS

**DNABERT**    Arguably the first DNA LM, DNABERT proposed in Ji et al. (2021) applies the BERT architecture from Devlin et al. (2018), with a few modifications, to genomic sequences. The authors train on the human genome and use $k$-mer tokens generated with sliding windows. Input sequences were 512 tokens, and the model was trained using the MLM objective, but with the restriction that masking was performed for contiguous tokens within a sequence. The downstream tasks focused on genome annotation, with promoter, transcription factor binding sites, and splice site classification. Of note, although DNABERT was pre-trained on human genome, it was fine-tuned on mouse downstream tasks as well, yielding competitive performance relative to supervised learning baselines.

**Nucleotide Transformer**    Following the success of model scaling in other domains, Dalla-Torre et al. (2023) explore scaling DNA foundation models in introducing the Nucleotide Transformer. They explore various model sizes – ranging from 500 million parameters to 2.5 billion, in their first generation release, and 50 million to 500 million parameters in their subsequent version 2 – and various pre-training data setups, including human reference genome, 3,000 diverse human genomes, and 850 multi-species reference genomes. They utilize non-overlapping 6-mer tokenization and a BERT-style architecture trained with an MLM objective. Other notable differences between the first and second version is that in version 2 input context size was scaled from 1,000 tokens to 2,000 and positional embeddings used in version 1 were learned whereas version 2 used rotary embeddings (Su et al., 2021), which have been shown to better extend to longer contexts. This work also introduced the Nucleotide Transformer suite of tasks, described in more detail below.

**DNABERT-2**    Building on the initial success of DNABERT, Zhou et al. (2023) present a model trained on multi-species genomes: 135 species, across 7 categories. They also change tokenization to byte-pair-encoding (Kudo and Richardson, 2018; Sennrich et al., 2015), with a vocabulary size of 4,096, arguing that overlapping $k$-mer tokenization makes the MLM task 'too easy' by leaking

information across tokens and that non-overlapping $k$-mer tokenization suffers from the drawback that minor changes to the input sequence, e.g., removing the first character, lead to drastically different tokenization outputs. They use input sequence lengths of 128 tokens. Additionally, Zhou et al. (2023) replace the learned positional embeddings from DNABERT with ALiBi (Press et al., 2022). DNABERT-2 was evaluated on a suite of downstream tasks introduced in Zhou et al. (2023) known as the Genome Understanding Evaluation (GUE).

**DNABERT-S**    DNABERT-S then further builds upon DNABERT-2 to generate richer DNA embeddings for uses in meta genomics. For DNABERT-S, Zhou et al. (2024) takes a trained DNABERT-2 model and they train it with a contrastive learning objective with a novel strategy with a new training dataset: 2 million paired DNA sequences from fungi, viruses, and bacteria. Additionally, their constrative learning approach involves a new strategy: Curriculum Contrastive Learning ($C^2$LR) that defines a training curriculum for the model and makes use of a new training objective: Manifold Instance Mixup (MI-Mix). Zhou et al. (2024) use this MI-Mix objective in addition to the the SimCLR contrastive loss objective for different phases of their training (Chen et al., 2020).

**HyenaDNA**    In contrast to the other language models reviewed above, the HyenaDNA model from Nguyen et al. (2023) is a next token prediction, uni-directional model. Using character-level tokenization and the Hyena layers (Poli et al., 2023) as a backbone, Nguyen et al. (2023) also propose a training recipe for scaling input context sizes up to 1 million bps. To evaluate their model they use a combination of downstream tasks, including the suite of tasks from Nucleotide Transformer (Dalla-Torre et al., 2023), a set of mouse and human genome annotation tasks presented in Grešová et al. (2023), the chromatin profiling tasks from DeepSea (Zhou and Troyanskaya, 2015), and a species classification task, where the model takes in sequences of various species and needs to output the correct species label.

**Evo2**    Evo (Nguyen et al., 2024) and it's successor Evo2 (Brixi et al., 2025) share a similar architecture to Hyena named StripedHyena, which is a hybrid between grouped attention layers and standard Hyena layers. These models also use character-level tokenization and are trained using next token prediction. Similar to Hyena, these models were extended to up to 1 Million base-pair inputs. The main difference between these two models is the dataset, as Evo1 was trained mainly on prokaryote sequences while Evo2 contained sequences from many diverse species, including various plants, vertebrates, and bacteria. Both models contain configurations of 7 billion parmaters, with Evo2 also releasing a 40B parameter variant.

In this work we only evaluate Evo2 due to our focus on human benchmarks, and we additionally only perform zero-shot evaluation due to the relatively high computational cost of running Evo2.

**Caduceus**    In the recent **Mamba** work (Gu and Dao, 2023), the authors pre-train various sized models that use the Mamba backbone on the human reference genome. Similar to HyenaDNA , the pre-training objective is next token prediction, tokenization is by nucleotide base, and input sequences are scaled up to 1 million bps. Building off this work, Schiff et al. (2024) introduced **Caduceus**, a bi-directional Mamba-based model that contains reverse complement equivariance inductive biases, demonstrating state-of-the art performance on several tasks, including several Nucleotide Transformer tasks (Dalla-Torre et al., 2023) and the Genomic Benchmark (Grešová et al., 2023).

**GPN-MSA**    GPN-MSA (Benegas et al., 2023a) is an instance of an **alignment-based** DNA language model. Instead of having a single sequence as an input, a Multiple Sequence Alignment (MSA) is used. This alignment contains the same human DNA sequences as other DNA LMs, but it additionally contains an MSA containing the sequence information corresponding to the human sequences from multiple (89) other species. This extra information allows the GPN-MSA model to model the human sequence conditioned on the evolutionary information provided by the species included in the MSA. GPN-MSA is based on the RoFormer (Su et al., 2021) architecture with the difference that it flattens and encodes the provided MSA into the hidden dimension of the embedding. GPN-MSA uses a single-nucleotide tokneizer and is trained with a context size of 128 bps.

GPN-MSA also has many notable differences from the remaining DNA LMs regarding training since GPN-MSA is trained on a curated subset of the human genome. Briefly, phastCONS (Siepel et al., 2005) is used to tag each nucleotide with a conservation probability, then the genome is subdivided

into windows and the 5% windows that are predicted to be the most highly conserved are selected for training, along with 0.1% of the remaining windows and a reverse complement of all the windows. GPN-MSA is then trained in a manner similar to the other MLM models, 15% of the human input sequences are masked and a weighted cross entropy loss is used for the reconstructed tokens. The weights for the Cross Entropy Loss attempt to downweight repetitive regions and upweight conserved regions, with the weights for each nucleotide being determined based on calling repeat regions and the phlyoP and phastCONS score (Siepel et al., 2005; Pollard et al., 2010) for that nucleotide.

Variant effect prediction takes the same form as for the other MLM models, except that GPN-MSA additionally takes as input the MSA for 89 other species at that location as well. Notably, the input MSA to GPN-MSA is never masked.

GPN-MSA shows very strong performance on variant effect prediction tasks, where having access to explicity evolutionary information in the form of an MSA proves a significant advantage. In theory, any sufficiently large DNA LM trained on the same species that GPN-MSA inclucedes in it's MSA should be able to match the performance of GPN-MSA (at least in the VEP tasks), providing a concrete target that existing DNA LMs should be aiming towards. The addition of an MSA does not solve all problems though, as GPN-MSAs performance lags behind in the reamining tasks, likely due to a combination of a relatively short context window and the decreased importance of modeling conservation of tasks like Bulk RNA that also require learning information apart from the sequence (such as the relationship between regulatory elements and tissues).

**Other DNA LMs**   While the models above represent those that we initially validate on our benchmark, the field of DNA LMs is growing at a rapid pace and consists of several notable works that we briefly describe below.

While not developed specifically as a DNA LM, the **BigBird** architecture proposed in Zaheer et al. (2020) was applied to genomic sequences to demonstrate its usefulness in long context tasks. Using sparse attention to reduce computational complexity of transformer (Vaswani et al., 2017) blocks from quadratic to linear, BigBird is able to effectively scale up to longer contexts. In Fishman et al. (2023), the authors present a family of foundation models, **GENA-LM**, aimed specifically at modeling longer DNA sequences. Pre-training with an MLM objective on human and multi-species genomes, they use BPE with a vocabulary size of 32,000. The backbone architectures are either BERT (Devlin et al., 2018) or BigBird (Zaheer et al., 2020), allowing them to extend input lengths up to 36k bps.

Focusing on plant genomes, Benegas et al. (2023b) pre-train a MLM model on unaligned reference genomes of the *Arabidopsis thaliana* species and seven related species within the Brassicales order. Using character-level tokenization they use input lengths of 512 bps with dilated convolutions to create their **GPN** model. With 25 layers, despite the relatively short training sequences, GPN can theoretically extend to sequence inputs of millions of bps.

### A.3   EXISTING DNA LANGUAGE MODEL BENCHMARKS

Existing benchmarks vary in several aspects, including the species considered, the specific tasks of interest, the framing of these tasks, and the evaluation methodologies employed. These proposed benchmarks include the Nucleotide Transformer Benchmark (Dalla-Torre et al., 2023), Genomic Benchmarks (Grešová et al., 2023), Genome Understanding Evaluation (GUE, (Zhou et al., 2023)), and Benchmarking DNA Language Models on Biologically Meaningful Tasks (BEND; Marin et al. (2023)).

Existing DNA benchmarks are primarily composed of classification tasks for sequence-wise predictions, ranging from cis-regulatory elements and splice sites to chromatin features and variant effects. These benchmarks not only compile and build datasets but also carry out evaluations of DNA LMs using both fine-tuning methods, where pre-trained models are trained in a supervised manner on the downstream tasks, and zero-shot prediction, where models are evaluated in their pre-trained state without additional fine-tuning.

**Nucleotide Transformer Benchmark**   Dalla-Torre et al. (2023) compile a set of 18 distinct genomic datasets framed as sequence-wise classification tasks. These tasks included 10 datasets related to epigenetic mark prediction in yeast genomes, three tasks predicting the presence of promoters in mouse and human genomes, two tasks predicting enhancer presence and activity levels in the human

genome, and three tasks predicting splice sites in multiple diverse species. Sequence lengths in this benchmark ranged from 200 to 600 bps. Additionally, the authors evaluated a set of DNA LMs and a supervised genomic model, Enformer (Avsec et al., 2021a), by fine-tuning these models on their benchmark using a robust 10-fold cross-validation protocol. Parameter-efficient fine-tuning methods with a classification head were used for Enformer, DNABERT, and NT models, while full fine-tuning with a classification head was applied to the HyenaDNA models. Limitations of this benchmark include the focus on short-range contexts, the inclusion of synthetic sequences as negative examples, and limited supervised baselines.

**Genomic Benchmarks**   Genomic Benchmarks (Grešová et al., 2023) is a collection of datasets for genomic sequence classification, composed of existing datasets and novel ones scraped from publicly available databases. The benchmark includes nine tasks focusing on regulatory element prediction, such as promoters, enhancers, and open chromatin regions. These tasks cover human, mouse, roundworm, and fly genomes, with average sequence lengths ranging from 200 to 2,370 bps. The authors also provide code to train simple convolutional network that can be used as a baseline. Similar to the Nucleotide Transformer benchmark, this benchmark focuses on short-range tasks, does not present a robust set of baselines, and contains potentially less impactful tasks, e.g., distinguishing between human and worm genomic sequences.

**Genomic Understanding Evaluation (GUE)**   The authors of the DNABERT-2 (Zhou et al., 2023) introduced the Genomic Understanding Evaluation (GUE) benchmark, which is divided into two groups by sequence length: GUE and GUE+. This benchmark comprises seven classification tasks, such as cis-regulatory element prediction and species classification, built from 28 datasets from multiple species. The inclusion of multiple species allows for the assessment of DNA LMs' generalizability. The tasks are curated to be appropriately challenging, including measures such as class balancing, adversarial sample inclusion, and reduction of training sample volume. GUE features sequence lengths ranging from 70 to 1k bps, while GUE+ includes sequence lengths from 5k to 10k bps. GUE evaluated DNABERT1 and 2, NT, and HyenaDNA models on their benchmark. HyenaDNA models are fully fine-tuned while DNABERT and NT models are fine-tuned using parameter efficient methods. The GUE benchmark results are limited since they do not cover a robust set of baselines but rather only present the simple supervised convolutional network from the Genomic Benchamark (Grešová et al., 2023). Additionally, only binary or multi-class sequence-wise classification tasks are considered and tasks of biological importance, such as variant effect prediction and gene expression are not included.

**Benchmarking DNA LLMs on Biologically Meaningful Tasks (BEND)**   BEND (Marin et al., 2023) is a recently proposed benchmark focused on compiling tasks that capture the complexity and intricacies of real-world genomic analysis. The authors collected seven different datasets, all from the human genome, covering gene finding, enhancer annotation, chromatin accessibility, histone modification, CpG methylation, and two types of variant effect prediction. Unlike previous benchmarks that focused solely on sequence-wise classification tasks, BEND also includes the task "Gene finding", which tests nucleotide-resolution modeling. In five out of seven tasks the input length is 512 bps, as these tasks are considered short-range. "Gene finding" task use sequences up to 14k bps. Their "Enhancer annotation" task uses 100k bp sequences, but it only contains 285 input sequences. Notably, for tasks in BEND that overlap with our benchmark (such as variant effect prediction), BEND uses a fixed context length of 512 bp, thus not evaluating the importance of extended context and variant-gene distal interactions on this type of task. Therefore, this benchmark is mostly limited to short-range tasks and does not include gene expression, an important and challenging task in genomics. This benchmark however makes progress in including a broader set of supervised methods as baselines. Unlike our work, models are only evaluated using partial fine-tuning, where backbone DNA LM weights are frozen for downstream task training.

**GenBench**   The GenBench suite (Liu et al., 2024) is composed of 43 different datasets split between "short" and "long" range tasks, where long-range tasks are defined by having a sequence length of greater than 1000 base pairs. The tasks in GenBench, spanning multiple species, are primarily binary, sequence-level classification tasks but also include multi-class classification and regression tasks. The authors evaluate six different genomic language models covering both attention and convolution-based architectures. While GenBench provides a comprehensive evaluation, it lacks

critical tasks like variant effect prediction in non-coding regions and zero-shot evaluations. It also omits comparisons to long-context models like Enformer and is limited in its evaluation of long-range tasks, with the longest sequence length capped at 30,000 base pairs.

**BEACON**  The BEACON benchmark (Ren et al., 2024) introduces the first unified evaluation framework for RNA modeling, encompassing 13 tasks across structural analysis, functional studies, and engineering applications. It evaluates 29 models, ranging from pre-trained RNA language models to naive supervised models, and examines the influence of tokenization strategies and positional embeddings on performance. While BEACON is a valuable resource for assessing RNA-focused models, its scope is distinct from genomic benchmarks, as it targets RNA-specific tasks rather than genomic applications like regulatory element prediction, variant effect prediction, or gene expression prediction.

# B  ADDITIONAL DETAILS ABOUT GENOMIC LONG RANGE BENCHMARK

We note that our datasets do not contain any personally identifiable information or offensive content.

Table 6 provides details describing the evaluation method used, dataset sizes, metric, and data sources. Additional details on task specific data curation and processing are described in the following subsections.

Table 6: Additional information for Genomic LRB tasks, including number of samples in train and test splits, metric, and data source.

| Task | Eval | Test split | Metric | Data Source |
|---|---|---|---|---|
| *Variant Effect Prediction* | | | | |
| Causal eQTL | Fine-tune & Zero-shot | Chromosome 9, 10 | AUROC | GTEx (via Avsec et al. (2021b)) |
| Pathogenic OMIM | Zero-shot | - | AUPRC | OMIM, gnomAD (via Benegas et al. (2023a)) |
| Pathogenic ClinVar | Fine-tune & Zero-shot | Chromosome 8 | AUROC | ClinVar, gnomAD (via Benegas et al. (2023a)) |
| *Gene Expression Prediction* | | | | |
| Bulk RNA Expression | Fine-tune | Chromosome 8 | $R^2$ | GTEx, FANTOM5 (via Zhou et al. (2018a)) |
| CAGE | Fine-tune | Cluster then Random | $R^2$ | FANTOM5 (via Kelley (2020)) |
| *Regulatory Element Detection* | | | | |
| Promoter | Fine-tune | Chromosome 8, 9 | AUPRC | SCREEN |
| Enhancer | Fine-tune | Chromosome 8,9 | AUROC | SCREEN |
| *Chromatin Feature Identification* | | | | |
| Histone Marks | Fine-tune | Chromosome 8, 9 | AUPRC | ENCODE, Roadmap Epigenomics (via Zhou and Troyanskaya (2015)) |
| DNA Accessibility | Fine-tune | Chromosome 8, 9 | AUPRC | ENCODE, Roadmap Epigenomics (via Zhou and Troyanskaya (2015)) |

## B.1  VARIANT EFFECT PREDICTION

### B.1.1  CAUSAL EQTL

**Data Processing**  Processed data in the form of `vcf` files for positive and negative variants across 49 different tissue types were obtained from Avsec et al. (2021a). Fine-mapped GTEx (Consortium, 2020) eQTLs originate from Wang et al. (2021), while the negative matched set of variants comes from Avsec et al. (2021a). The statistical fine-mapping tool SuSiE (Wang et al., 2020) was used to label variants. Variants from the fine-mapped eQTL set were selected and given positive labels if their posterior inclusion probability was $> 0.9$, as assigned by SuSiE. Variants from the matched negative set were given negative labels if their posterior inclusion probability was $< 0.01$. DNA sequences were obtained from the human reference genome assembly GRCh38 (Schneider et al., 2017).

### B.1.2  PATHOGENIC OMIM

**Data Processing**  Processed data was obtained from Benegas et al. (2023a) in the form of `parquet` files with columns for SNP location, reference and alternative alleles, and pathogenicity label. Positive labeled data originates from a curated set of pathogenic variants located in the Online Mendelian Inheritance in Man (OMIM) (Smedley et al., 2016) catalog. The negative set is comprised of variants that are defined as common from gnomAD (Chen et al., 2022). gnomAD version `3.1.2` was downloaded and filtered to variants with allele number of at least 25,000. Common variants were defined as those with minor allele frequency (MAF) $> 5\%$. The input sequences were constructed

by selecting the appropriate genomic region from the human reference genome assembly GRCh38 (Schneider et al., 2017) and applying the changes specified by the given variants.

### B.1.3    PATHOGENIC CLINVAR

**Data Processing**  Processed data was obtained from Benegas et al. (2023a) in the form of `parquet` files with columns for SNP location, reference and alternative alleles, and pathogenicity label. Positive labels correspond to pathogenic variants originating from ClinVar (Landrum et al., 2020) whose review status was described as having at least a single submitted record with a classification but without assertion criteria. The negative set are variants that are defined as common from gnomAD (Chen et al., 2022). gnomAD version `3.1.2` was downloaded and filtered to variants with allele number of at least 25,000. Common variants were defined as those with MAF > 5%. Sequences were obtained from the human reference genome assembly GRCh38 (Schneider et al., 2017).

**Short-Range**  The ClinVar dataset is mostly variants in coding regions, and since most human protein sequences have less than 1,000 amino acids predicting the impact of coding variants should require orders of magnitude smaller context windows than non-coding variants. Therefore, we consider this task as potentially short-range.

## B.2    GENE EXPRESSION PREDICTION

### B.2.1    BULK RNA-SEQ

**Data Processing**  Processed data in the form `csv` files that contained gene TSS locations, strand, and RNA expression RPKM counts across 218 tissue types was obtained from ExPecto (Zhou et al., 2018a). Expression data originates from GTEx (Consortium, 2020), while representative TSS locations were determined in ExPecto. The authors of ExPecto determined representative TSS for Pol II transcribed genes based on quantification of CAGE reads from the FANTOM5 project (Forrest et al., 2014). The specific procedure they used is as follows, a CAGE peak was associated to a GENCODE (Harrow et al., 2012) gene if it was withing 1000 bps from a GENCODE `v24` annotated TSS. The most abundant CAGE peak for each gene was then selected as the representative TSS. When no CAGE peak could be assigned to a gene, the annotated gene start position was used as the representative TSS. We $\log(1 + x)$ normalized then standardized the RNA-seq counts before training models. Sequences centered around the TSS were obtained from the human reference genome assembly GRCh37 (Church et al., 2011).

### B.2.2    CAP ANALYSIS GENE EXPRESSION (CAGE) PROFILE

**Data Processing**  Processed data was obtained from Basenji2 (Kelley, 2020), where input sequence locations were collected as `bed` files and CAGE counts as `TensorFlow` (Abadi et al., 2015) records. Original data comes from the FANTOM5 project (Forrest et al., 2014). Data was processed to produce CAGE labels for non-overlapping 128 bp bins within a sequence of 114,688 bps. For each bin, there are 638 different predictions corresponding to the CAGE count in various cell, tissue, or treatment types (e.g., fibroblast, heart, or monocytes treated with Salmonella). This resulted in an output array of 896 bins × 638 tracks for a single sample. DNA sequences were obtained from the human reference genome assembly GRCh38 (Schneider et al., 2017).

The compute requirements to store and process this data make it more difficult and less accessible to users. To achieve a balance of user-friendliness while also maintaining a representative view of the data, we sub-sampled the number of tracks to 50 by using the following guidelines:

1. Only select one cell line.
2. Only keep mock treated and remove other treatments.
3. Only select one donor.

The 50 specific tracks which were selected can be found in Table 7 below. This maintains the number of sequences in the entire dataset but reduces the number of labels for each sequence from 638 to 50 thus reducing storage requirements from ∼84GB to ∼7GB.

Table 7: The 50 CAGE tracks sub-sampled for the Genomic LRB from the original 638 tracks.

| Track Index | Description |
| --- | --- |
| 0 | CAGE:adipose tissue, adult, pool1 |
| 1 | CAGE:bladder, adult, pool1 |
| 2 | CAGE:brain, adult, pool1 |
| 3 | CAGE:cervix, adult, pool1 |
| 4 | CAGE:colon, adult, pool1 |
| 5 | CAGE:esophagus, adult, pool1 |
| 6 | CAGE:heart, adult, pool1 |
| 7 | CAGE:kidney, adult, pool1 |
| 8 | CAGE:liver, adult, pool1 |
| 9 | CAGE:lung, adult, pool1 |
| 10 | CAGE:ovary, adult, pool1 |
| 11 | CAGE:placenta, adult, pool1 |
| 12 | CAGE:prostate, adult, pool1 |
| 13 | CAGE:skeletal muscle, adult, pool1 |
| 14 | CAGE:small intestine, adult, pool1 |
| 15 | CAGE:spleen, adult, pool1 |
| 16 | CAGE:testis, adult, pool1 |
| 17 | CAGE:thymus, adult, pool1 |
| 18 | CAGE:thyroid, adult, pool1 |
| 19 | CAGE:trachea, adult, pool1 |
| 20 | CAGE:retina, adult, pool1 |
| 21 | CAGE:temporal lobe, adult, pool1 |
| 22 | CAGE:postcentral gyrus, adult, pool1 |
| 23 | CAGE:pons, adult, pool1 |
| 24 | CAGE:parietal lobe, adult, pool1 |
| 25 | CAGE:paracentral gyrus, adult, pool1 |
| 26 | CAGE:occipital pole, adult, pool1 |
| 27 | CAGE:nucleus accumbens, adult, pool1 |
| 28 | CAGE:medulla oblongata, adult, pool1 |
| 29 | CAGE:insula, adult, pool1 |
| 30 | CAGE:frontal lobe, adult, pool1 |
| 31 | CAGE:dura mater, adult, |
| 32 | CAGE:corpus callosum, adult, pool1 |
| 33 | CAGE:adenocarcinoma cell line:IM95m |
| 34 | CAGE:breast carcinoma cell line:MCF7 |
| 35 | CAGE:diffuse large B-cell lymphoma cell line:CTB-1 |
| 36 | CAGE:glioma cell line:GI-1 |
| 37 | CAGE:liposarcoma cell line:SW 872 |
| 38 | CAGE:Sebocyte, |
| 39 | CAGE:CD4+ T Cells, |
| 40 | CAGE:Natural Killer Cells, |
| 41 | CAGE:Neutrophils, |
| 42 | CAGE:Pericytes, |
| 43 | CAGE:Alveolar Epithelial Cells, |
| 44 | CAGE:Renal Mesangial Cells, |
| 45 | CAGE:Nucleus Pulposus Cell, |
| 46 | CAGE:Keratocytes, |
| 47 | CAGE:Mesenchymal Stem Cells - adipose, |
| 48 | CAGE:Mammary Epithelial Cell, |
| 49 | CAGE:Osteoblast, |

### B.3 CIS-REGULATORY ELEMENT DETECTION

**Data Processing** Original data was sourced from Search Candidate cis-Regulatory Elements v3 (SCREEN) registry by ENCODE (Moore et al., 2020). The data is processed as follows, we break the human reference genome into 200 bp non-overlapping chunks. If the 200 bp chunk overlaps by at least 50% or more with a contiguous region from the set of annotated cis-regulatory elements (promoters or enhancers), we label them as positive, else the chunk is labeled as negative. The resulting dataset was composed of ∼15M negative samples and ∼50k positive promoter samples and ∼1M positive enhancer samples We randomly sub-sampled the negative set to 1M samples, and kept all positive samples, to make this dataset more manageable in size. DNA sequences were obtained from the human reference genome assembly GRCh38 (Schneider et al., 2017).

**Short-Range** Since this task involves predicting the presence of a regulatory element within a specific sequence, only local context is believed to be important. The activity of promoters and enhancers in different cell types is dictated by the presence of binding sites for specific proteins (Andersson and Sandelin, 2020) and thus likely do not require long-distance interactions, as demonstrated by the high predictive value of models using less than 1k bp input sequences (Avsec et al., 2021b; Kelley et al., 2016).

### B.4 CHROMATIN FEATURE IDENTIFICATION

**Data Processing** Processed data was obtained from DeepSea (Zhou and Troyanskaya, 2015) in the form of 1k bp sequences and labels as txt files. Original chromatin profiling data comes from ENCODE and Roadmap Epigenomics (Moore et al., 2020; Bernstein et al., 2010). The authors of DeepSea processed the data by chunking the human genome into 200 bp bins where for each bin labels were determined for hundreds of different chromatin features. Only bins with at least one transcription factor binding event were considered for the dataset. If the bin overlapped with a peak region of the specific chromatin profile by more than half of the sequence, a positive label was assigned. DNA sequences were obtained from the human reference genome assembly GRCh37 (Church et al., 2011). To make the dataset more accessible, we randomly sub-sampled the chromatin profiles from 125 to 20 tracks for the histones dataset and from 104 to 20 tracks for the DNase dataset. The sub-sampled tracks for both datasets can be found in Table 8 and Section B.4.

**Short-Range** Chromatin features are not expected to be strongly influenced by long-range interactions. Most of the information affecting these chromatin features occurs locally and depends on the binding of different proteins (Lee and Young, 2013). This is also corroborated by the high predictive value of models using less than 1k bps input sequences (Kelley et al., 2016; Zhou and Troyanskaya, 2015).

Table 8: 20 Histone tracks sub sampled for the Genomic LRB from the original 104 tracks with histone mark and cell type information.

| Track Index | Histone Mark | Cell Type |
| --- | --- | --- |
| 0 | H2BK12ac | H1-hESC |
| 1 | H3K4me1 | NHEK |
| 2 | H3K4me2 | NH-A |
| 3 | H3K9me1 | K562 |
| 4 | H4K20me1 | NHEK |
| 5 | H2BK5ac | H1-hESC |
| 6 | H3K4me3 | NH-A |
| 7 | H4K8ac | H1-hESC |
| 8 | H3K4me2 | Monocytes-CD14+RO01746 |
| 9 | H3K27me3 | Osteoblasts |
| 10 | H3K36me3 | Monocytes-CD14+RO01746 |
| 11 | H3K23me2 | H1-hESC |
| 12 | H3K27ac | NHLF |
| 13 | H3K36me3 | NHEK |
| 14 | H2BK20ac | H1-hESC |
| 15 | H3K9ac | NHLF |
| 16 | H3K36me3 | Osteoblasts |
| 17 | H2BK120ac | H1-hESC |
| 18 | H3K79me2 | K562 |
| 19 | H3K4me1 | K562 |

Table 9: 20 DNase tracks sub sampled for the Genomic LRB from the original 125 tracks with cell type and treatment information.

| Track Index | Treatment | Cell Type |
|:---:|:---:|:---:|
| 0 | None | SAEC |
| 1 | None | HRPEpiC |
| 2 | None | SK-N-MC |
| 3 | None | RWPE1 |
| 4 | None | Th2 |
| 5 | None | Adult_CD4_Th0 |
| 6 | None | HMEC |
| 7 | None | NHEK |
| 8 | UT189 | Urothelia |
| 9 | None | pHTE |
| 10 | None | Urothelia |
| 11 | None | WERI-Rb-1 |
| 12 | None | Huh-7 |
| 13 | None | A549 |
| 14 | None | Th1 |
| 15 | None | HA-h |
| 16 | None | RPTEC |
| 17 | None | HMVEC-dBl-Ad |
| 18 | None | HGF |
| 19 | None | HMF |

### B.5 VISUALIZATION TOOL

The annotations that we join to our task datasets come from the human reference genome assembly GRCh38 (Schneider et al., 2017). To obtain these annotation we follow the methodology reported in SegmentNT (de Almeida et al., 2024) for data curation. Annotations include genomic elements, such as enhancers, exon, intron, 5' UTR, etc. The location of all gene elements and polyA signals were obtained from GENCODE (v44) (Harrow et al., 2012) gene annotation. Promoter, enhancer, and CTCF-bound sites were retrieved from ENCODE's SCREEN database (The ENCODE Project Consortium, 2020). Promoters and enhancers were split into tissue-invariant and tissue-specific annotations, following the tissue-invariant annotations from Meuleman et al. (2020). Briefly, if a promoter or enhancer overlapped at all with a region annotated as tissue-invariant, that promoter or enhancer was annotated as tissue-invariant. All other promoters and enhancers were tagged as tissue specific. Scripts from HISAT2 (Kim et al., 2019) were used to extract respective intron and splice site annotations. Annotations of repeat regions were collected from RepeatMasker (Smit et al., 2015).

Annotations were merged into the dataset by aligning chromosome and regions (start / stop position) of annotations with the genomic locations associated with the compiled tasks in the Genomics LRB. That is, if the sequence positions in our dataset overlapped with regions in the annotation files, the sequence was tagged with the corresponding annotation. For example, for variant effect prediction tasks, the SNP location was used for the merge; for regulatory element detection tasks, the start and stop positions were used. Specifically, a sample in our dataset was associated with an annotation if the sample position was both greater than the starting position of the annotation and less than the ending position of the annotation.

The UCSC liftover browser tool (Hinrichs et al., 2006) was used to convert GRCh38 annotations to the GRCh37 reference assembly locations to be associated with datasets relying on GRCh37 locations.

With annotations merged into the datasets in our Genomics LRB, we develop a visualization tool that enables users to 'slice' results. Our tool is an interactive `jupyter` (Kluyver et al., 2016) notebook that enables toggling different models and has visualizations for aggregate results, results by distance to nearest TSS / enhancer, and results by annotation. In Figure 4, we provide selected screenshots from our visualisation tool demonstrating how a user can view results for each task, select different models, and split by various annotations.

### B.6 ARBITRARY SEQUENCE LENGTH

To enable users to download arbitrarily long sequence lengths, samples for each task are stored either as single positions in the genome (e.g., the SNP location for variant effect prediction or the TSS for bulk RNA expression) or as start and stop locations for tasks like regulatory element and chromatin feature prediction. In addition we store the human reference genome assemblies GRCH38 (Schneider et al., 2017) and GRCH37 (Church et al., 2011). The `PyFaidx` Python package (Shirley et al., 2015) is used to create an indexed `FASTA` file object from the reference genomes for fast random access to any subsequence. With the user's requested sequence length, we symmetrically extend sequence locations from our datasets and use these extended indices to extract the underlying DNA sequence from the indexed reference genomes. If the extended sequence indexes beyond a chromosome boundary, the sample is not returned.

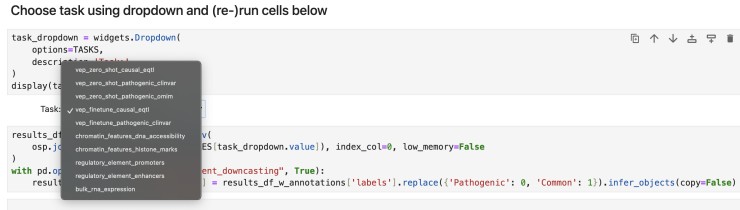

(a) Screenshot of the visualization tool showing the ability to select different tasks from the Genomics LRB.

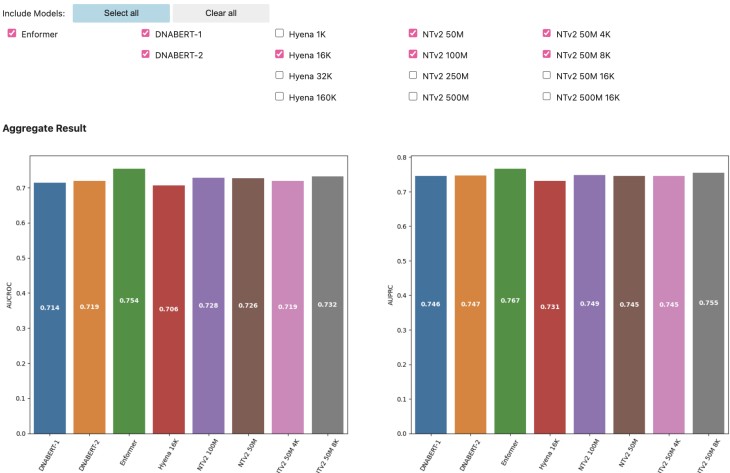

(b) Screenshot of the visualization tool showing the ability to select different models for comparison.

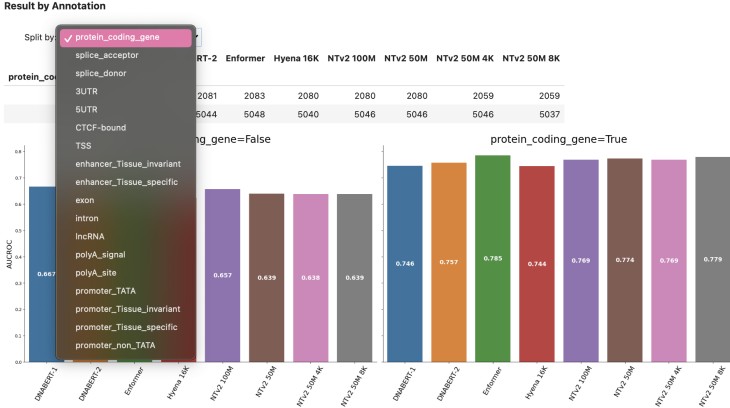

(c) Screenshot of the visualization tool showing the ability to select different annotations by which to split results.

Figure 4: Sample screenshots from our interactive visualization tool.

## C  CONTEXT LENGTH EXTENSION

Motivated by the long-range sequences present in the LRB, we explore methods for extending the context size of existing models. To that end, we focus on the Nucleotide Transformer model (NTv2; Dalla-Torre et al. (2023)), which originally has a context size of 12k bp and uses rotary positional embeddings (RoPE; Su et al. (2021)). However, processing longer sequences with LMs like NTv2, which use the transformer architecture (Vaswani et al., 2017), faces two main challenges. First, transformers rely on the attention mechanism, which scales quadratically in sequence length. Second,

LMs struggle with generalizing to sequence lengths beyond those seen during pre-training, known as length extrapolation (Anil et al., 2022; Dubois et al., 2019; Kazemnejad et al., 2023; Press et al., 2021).

**Methodology**   To address the compute constraints, we use a memory-efficient attention implementation, computing attention scores sequentially and in chunks of $\sqrt{L}$, reducing memory usage from $\mathcal{O}(L^2)$ to $\mathcal{O}(\sqrt{L})$, where $L$ denotes sequence length (Rabe and Staats, 2021). To solve the length generalization issue, we apply the 'NTK-aware' method presented in Peng et al. (2023). This method re-scales the frequencies in RoPE embeddings to handle longer sequences by converting length extrapolation into *interpolation*.

**Rotary Embeddings**   In attention-based modules, such as those used in transformer models (Vaswani et al., 2017), for a sequence of length $L$, the model takes embeddings in $\{\mathbf{x}\}_{j=1}^{L}, \mathbf{x}_j \in \mathbb{R}^d$, where $d$ is the dimension of the embeddings, and computes query, key, and value vectors at every $m^{\text{th}}$ and $n^{\text{th}}$ position in the sequence:

$$\mathbf{q}_m = f_q(\mathbf{x}_m, m)$$
$$\mathbf{k}_n = f_k(\mathbf{x}_n, n)$$
$$\mathbf{v}_n = f_v(\mathbf{x}_n, n).$$

$f_q, f_k, f_v$ are query, key, and value transformations, respectively. For rotary embeddings (RoPE (Su et al., 2021)), we can think of $\mathbb{R}^d$ as equivalent to the complex field $\mathbb{C}^{d/2}$ and define $f_q$ and $f_k$ as:

$$f_q(\mathbf{x}_m, m) = e^{im\Theta}\mathbf{W}_q\mathbf{x}_m$$
$$f_k(\mathbf{x}_n, n) = e^{in\Theta}\mathbf{W}_k\mathbf{x}_n,$$

where $\mathbf{W}_q$ and $\mathbf{W}_k$ are linear transformations and $\Theta = \text{diag}(\theta_1, \ldots, \theta_{d/2})$ is a diagonal matrix, with $\theta_j = b^{-2j/d}$ and $b = 10000$.

**RoPE Position Interpolation**   In the concurrent works of Chen et al. (2023) and kaiokendev (2023), the method of position interpolation was introduced, whereby longer sequences of length $L' > L$ are accommodated by simply rescaling the position input to $f_q$ and $f_k$, e.g., $f_q(\mathbf{x}_m, m\frac{L}{L'})$.

**NTK-aware RoPE Interpolation**   An alternative interpolation scheme, attributed to bloc97 (2023), is motivated by the hypothesis that position interpolation may lead to the loss of high frequency information. The approach that purportedly resolves this issue is related to the theory of Neural Tangent Kernels (NTK) by means of an analogy between RoPE and Fourier Features (Tancik et al., 2020), and is thus named "NTK-aware" interpolation. This scheme is characterized by a rescaling applied not to the position but rather to the basis of rotation, as follows:

$$\theta_j = b'^{-2j/d}$$
$$b' = b \cdot \left(\frac{L}{L'}\right)^{\frac{d}{d-2}}$$

In the experiments on context extension presented in the main text, we adopt this interpolation scheme.

We note that the authors in Peng et al. (2023) further tweak and build on NTK-aware interpolation to create their proposed interpolation scheme, which they title YaRN. However, the full YaRN approach, as presented in Peng et al. (2023) requires several manually tuned hyperparameters, which were carefully selected for the decoder-only generative Llama-2 7 billion parameter model (Touvron et al., 2023a;b). We therefore adopted the simpler NTK-aware approach in our experiments.

**Efficient Long-Range Context Extension**   To mitigate the computational and memory costs of scaling to larger contexts, we follow the algorithm presented in Rabe and Staats (2021). This algorithm leverages a "lazy softmax" approach where key-value pairs are processed sequentially, maintaining only two vectors in memory: one for the accumulated weighted values and another for the cumulative sum of weights. This method significantly reduces memory usage by avoiding the storage of all pairwise attention scores. To optimize performance on modern hardware accelerators, which rely on parallelization for efficiency, the implementation processes attention in chunks. Rabe and

# D  ADDITIONAL EXPERIMENTAL DETAILS

## D.1  EVALUATED DNA LANGUAGE MODELS

In Table 10, we list the DNA LMs included in the initial evaluation of our benchmark.

Table 10: Overview of Pre-trained DNA LMs evaluated in this study.

| | Pre-training | Data | Parameters | Architecture | Context (bps) | Tokenization |
|---|---|---|---|---|---|---|
| NTv2 | MLM | Multi-Species | 50M, 100M, 250M, 500M | Transformer | 12k | 6-mer |
| DNABERT-1 | MLM | Human Reference | 88.6M | Transformer | 512 bps | 6-mer |
| DNABERT-2 | MLM | Multi-Species | 116.6M | Transformer | 700 (train), up to 10k (eval) | Byte Pair Encoding |
| HyenaDNA | NTP | Human Reference | 1.6M, 2.6M, 3.9M,12.9M | SSM | 1k, 16k, 32k, 160k | Single Base Pair |
| Caduceus | MLM | Human Reference | 7.7M | SSM | 131k | Single Base Pair |
| Evo2 | NTP | Multi-Species | 1B, 7B, 40B | Hybrid | 8k, 1M | Single Base Pair |
| GPN-MSA | MLM | Human Reference + Multi-Species MSA | 86M | Transformer | 128 | Single Base Pair |

## D.2  ZERO-SHOT EVALUATION

**MLM Models**  For masked DNA LMs, zero-shot scores are computed by masking the variant position in the sequence, performing inference on the masked sequence, and obtaining the probability distribution at the variant position. A score is then calculated using the probabilities of the reference allele token and the alternative allele token. For auto-regressive DNA LMs, no masking is required due to their unidirectional nature. Instead, a forward pass is done with the reference sequence, and the probability distribution is extracted from the token immediately preceding the variant position. For Alignment Models, the human sequence is treated in the same way as the other masked DNA LMs, but the auxiliary MSA is left entirely unmasked. Scores are computed as the log probability ratio for the reference (ref) and alternative (alt) allele tokens:

$$\text{variant effect score} = \log\left(\frac{P_{\text{ref}}}{P_{\text{alt}}}\right)$$

**Auto-regressive Models**  For auto-regressive models like Evo2, we follow the same variant scoring approach as in the original Evo2 work (Brixi et al., 2025). Briefly, instead of obtaining the probability distribution at the variant position as in MLM models, we compute the "psuedo log likelihood" (pLL) of the entire sequence centered around the variant of interest. Given an input sequence $x$ of length $L$ ($x \in \{A, T, C, T\}^L$) with reference nucleotide $x_{\text{ref}}$ or alternate nucleotide $x_{\text{alt}}$ at position $idx$, the variant score is calculated as:

$$\text{score} = PLL_{\text{alt}} - PLL_{\text{ref}}$$

$$PLL_{\text{ref}} = \sum_{i=0}^{idx-1} \log P(x_i|x_{0:i-1}) + \log P(x_{\text{ref}}|x_{0:idx-1}) + \sum_{j=idx+1}^{L} \log P(x_j|x_{0:idx-1}, x_{\text{ref}}, x_{idx+1:j-1})$$

$$PLL_{\text{alt}} = \sum_{i=0}^{idx-1} \log P(x_i|x_{0:i-1}) + \log P(x_{\text{alt}}|x_{0:idx-1}) + \sum_{j=idx+1}^{L} \log P(x_j|x_{0:idx-1}, x_{\text{alt}}, x_{idx+1:j-1})$$

Here $x_{i:j}$ denotes the sub sequence of $x$ starting at index $i$ and ending at index $j$ (inclusive). In practice, the Evo2 code base takes the mean PLL for a sequence instead of the sum, dividing the final variant effect score by $\frac{1}{L}$. Additionally, Evo2 defines their score as the reciprocal of the ratio used in MLM models. We follow both of these conventions when re-implementing their scoring methodology in this work.

Details about additional processing required for zero-shot prediction are given below.

### D.2.1 CAUSAL EQTL

The original dataset used for this tasks contains tissue information for each sequence. Given that zero-shot evaluate cannot account for tissue, we process variants appearing across multiple tissue types as follows: first, we find variants appearing in multiple tissues and determining a consensus label for a given variant across tissues using a 70% majority class agreement threshold. Variants appearing across multiple tissues whose majority class agreement was below this threshold were dropped. When computing metrics we only count variants appearing across tissues once.

### D.2.2 PATHOGENIC-OMIM

Due to computational considerations and given that this data set totals $\sim$2.3M examples, we only considered a subset of the common variants for carrying out zero-shot prediction. Specifically, we sub-sampled 200k common variants and kept all 406 original pathogenic variants.

## D.3 FINE-TUNING EVALUATION

To fine-tune models on our benchmark tasks, we first extracted model embeddings, in the case of DNA LMs this involves extracting the output of the last layer before the LM head, and in the case of Enformer, this involves extracting the model embeddings before the final supervised prediction head. Model embeddings were then processed in a task specific manner and subsequently fed into a task specific MLP, both of which are outlined below. We note that for Enformer, since it is a model that was originally trained in a multi-task supervised fashion and not intended to be fine-tuned, embeddings were frozen and only the prediction head was trained.

### D.3.1 CAUSAL EQTL

**Embedding Extraction** We extract model embeddings for both the reference and alternative sequences and average embeddings across a window of size 1536 bps symmetrically around the SNP position. The mean embeddings for the reference and alternative are concatenated. Tissue information is converted to one-hot and additionally concatenated to the reference-alternative embedding vector.

**MLP Head** MLP hidden dimensions are sized in an adaptive way such the hidden state size is equal to two times the base model's embedding dimension. The MLP is composed of one linear layer with size $2 \times$ embedding dimension, a softplus activation, another linear layer with size $2 \times$ embedding dimension, a softplus activation, and a final linear layer for binary prediction.

**Hyperparameters** The parameters used to fine-tune models on this task include batch size = 64, learning rate = $1e^{-5}$, ADAM (Kingma and Ba, 2014) optimizer with $\beta_1$= 0.9, $\beta_2$ = 0.999, and $\epsilon = 1e^{-8}$, trained for 1 epoch on the task's training dataset. Validation is carried out every 70 parameter update steps.

### D.3.2 PATHOGENIC CLINVAR

**Embedding Extraction** We extract model embeddings for both the reference and alternative sequences and take a window mean of size 1536 bps symmetrically around the SNP position. The mean embeddings for the reference and alternative are concatenated together.

**MLP Head** MLP hidden dimensions are sized in an adaptive way such the hidden state size is equal to two times the base model's embedding dimension. The MLP is composed of one linear layer with size $2 \times$ embedding dimension, a softplus activation, and a final linear layer for binary prediction.

**Hyperparameters** The parameters used to fine-tune models on this task include batch size = 64, learning rate = $1e^{-5}$, ADAM optimizer with $\beta_1$ = 0.9, $\beta_2$ = 0.999, and $\epsilon = 1e^{-8}$, trained for 3 epochs on the task's training dataset. Validation is carried out every 40 parameter update steps.

### D.3.3 BULK RNA EXPRESSION

**Embedding Extraction** We extract model embeddings for the input sequence and take perform mean pooling on a window centered on the TSS with 383 bps before the TSS and 256 bp after.

**MLP Head** MLP hidden dimensions are sized in an adaptive way such the hidden state size is equal to two times the base model's embedding dimension. The MLP is composed of one linear layer with size $2 \times$ embedding dimension, a softplus activation, and a final linear layer for predicting 218 regression values.

**Hyperparameters** The parameters used to fine-tune models on this task include batch size = 64, learning rate = $3\mathrm{e}^{-5}$, ADAM optimizer with $\beta_1 = 0.9$, $\beta_2 = 0.999$, and $\epsilon = 1\mathrm{e}^{-8}$ trained for 3 epochs on the task's training dataset. Validation is carried out every 50 parameter update steps.

### D.3.4    CAGE PREDICTION

**Embedding Extraction** Base model embeddings were extracted and fed into the task MLP predictor.

**MLP Head** MLP hidden dimensions are sized in an adaptive way such the hidden state size is equal to two times the base model's embedding dimension. The MLP is composed of one linear layer with size $2 \times$ embedding dimension, a softplus activation, and a final linear layer for predicting 50 regression values.

**Hyperparameters** The parameters used to fine-tune models on this task include batch size = 64, learning rate = $3\mathrm{e}^{-5}$, adam optimizer with $\beta_1 = 0.9$, $\beta_2 = 0.999$, and $\epsilon = 1\mathrm{e}^{-8}$ trained for 1 epoch of the training dataset. Validation is carried out every 50 parameter update steps.

### D.3.5    REGULATORY ELEMENTS

Due to computational considerations, we only fine-tuned models on a randomly sampled 100k subset of the full $\sim$1-2M samples in the training set . Models were evaluated on the full test dataset.

**Embedding Extraction** Given that the task is defined on predicting the presence of a regulatory element in the center 200 bp of the sequence, we extract a central window of 200 bps from the sequence of embeddings and perform mean pooling. This mean embedding is then passed as input to the MLP predictor head.

**MLP Head** MLP hidden dimensions are sized in an adaptive way such the hidden state size is equal to two times the base model's embedding dimension. The MLP is composed of one linear layer with size $2 \times$ embedding dimension, a softplus activation, and a final linear layer for predicting binary values.

**Hyperparameters** The parameters used to fine-tune models on this task include batch size = 64, learning rate = $3\mathrm{e}^{-5}$, ADAM optimizer with $\beta_1 = 0.9$, $\beta_2 = 0.999$, and $\epsilon = 1\mathrm{e}^{-8}$ trained for 1 epoch of the sampled training dataset for each task. Validation is carried out every 30 parameter update steps.

### D.3.6    CHROMATIN FEATURES

Due to computational considerations, we only fine-tuned models on a randomly sampled 100k subset from the full $\sim$2M sample training set. Models were evaluated on the full test dataset.

**Embedding Extraction** Given that the task is defined on predicting the presence of a chromatin feature in the center 200 bp of the sequence, we extract a central window of 200 bps from the sequence of embeddings and perform mean pooling. This mean embedding is then passed as input to the MLP predictor head.

**MLP Head** MLP hidden dimensions are sized in an adaptive way such the hidden state size is equal to two times the base model's embedding dimension. The MLP is composed of one linear layer with size $2 \times$ embedding dimension, a softplus activation, and a final linear layer for predicting the 20 binary labels.

**Hyperparameters** The parameters used to fine-tune models on this task include batch size = 64, learning rate = $3\mathrm{e}^{-5}$, adam optimizer with $\beta_1 = 0.9$, $\beta_2 = 0.999$, and $\epsilon = 1\mathrm{e}^{-8}$ trained for 1 epoch of the training dataset. Validation is carried out every 30 parameter update steps.

### D.4 FINE-TUNING ABLATION DETAILS

For the fine-tuning ablation study, we compared training only the task MLP with DNA LM embeddings frozen against training all DNA LM weights in conjunction with the task MLP. All training setup details regarding embedding extraction and hyperparameters were kept constant except for learning rate which was adjusted to account for training larger networks when full fine-tuning. The following learning rates for each task were used in the MLP only training:

- Variant effect prediction tasks: $1e^{-4}$
- Bulk RNA: $2.5e^{-4}$
- CAGE: $2e^{-4}$
- Regulatory elements: $2.5e^{-4}$
- Chromatin features: $2.5e^{-4}$.

### D.5 CONTEXT EXTENSION IMPLEMENTATION DETAILS

To conduct context length extension of NTv2, we first used the 50M model due to computation considerations. We started with the pre-trained NTv2 50M checkpoint from Dalla-Torre et al. (2023), pre-trained on 12k bp sequences, and extended the context length by factors of two to 24k, 48k, and 96k bps using a second stage of masked language modeling on a multi-species dataset from Dalla-Torre et al. (2023). After proving out this methodology for the 50M model, we conducted context length extension for the 500M model at 96k bps.

**Hyperparameters** For the 50M NTv2 model we use the following hyperparameters: batch size = 1M tokens, full precision training, masking ratio = 0.15, masking probability = 0.8, random token probability = 0.1. The ADAM optimizer with weight decay regularization was used with weight decay = 0.01, $\beta_1 = 0.9$, $\beta_2 = 0.999$, $\epsilon = 1e^{-8}$, a modified square decay learning rate schedule, with initial learning rate of $6e^{-5}$ and end learning rate of $8e^{-4}$ with 1000 warm up steps. Training was conducted over ∼5 billion tokens totalling ∼5k parameter update steps.

All hyperparameters were kept constant for the NTv2 500M model, however due to limited memory resources, mixed precision training was used.

### D.6 SUPERVISED TRAINING BASELINES DETAILS

**Convolutional Neural Network** The CNN architecture is comprised of eight 1D convolutional blocks that use a filter size of 5 and padding that keeps input sequence length unchanged and hidden dimension of 512. Each convolution block is composed of a convolutional layer followed by the GeLU non-linearity (Hendrycks and Gimpel, 2016) and a layer norm. After each convolutional block we also apply a fully-connected layer with GeLU non-linearity and another layer norm. Each block also contains a residual connection. This architecture is derived from the one used in Benegas et al. (2023b), but without dilation. The model consists of 12M parameters. The baseline model was trained with base-pair level tokenization and an input context size of 2,048 tokens.

**Caduceus** We also train an eight layer Caduceus (Schiff et al., 2024) model with hidden dimension of 256. We use the reverse complement equivariant version of this architecture (Caduceus-PS). The model consists of 3.3M parameters. The baseline model was trained with base-pair level tokenization and an input context size of 2,048 tokens.

## E ADDITIONAL RESULTS

### E.1 FULL DNA LM SERIES EVALUATIONS

In Tables 11 and 12 we display results for the full set of models evaluated on our benchmark.

DNABERT-2 and DNABERT-S were not fine-tuned on the CAGE task due to the incompatibility between the byte pair tokenization this model employs and binned labels used in this task. Additionally, given that DNABERT-S is trained on a contrastive learning objective and not a language

Table 11: Benchmarking performance of DNA LMs and baselines on variant effect prediction tasks. Models were evaluated in both fine-tuning and zero-shot settings. *Extended NTv2 500 M was fine-tuned with 60k bp sequences due to compute constraints. ** Caduceus 131k results omitted from the main table due to being a work in progress.

| Model Name | Context (bp) | Causal eQTL (AUROC) | | Pathogenic ClinVar (AUROC) | | Pathogenic OMIM (AUPRC) |
|---|---|---|---|---|---|---|
| | | Fine-tune | Zero-shot | Fine-tune | Zero-shot | Zero-shot |
| DNABERT 1 | 512 | $0.72 \pm 0.003$ | 0.51 | $0.67 \pm 0.037$ | 0.50 | 0.002 |
| DNABERT 2 | 10k | $0.72 \pm 0.008$ | 0.50 | $0.74 \pm 0.013$ | 0.50 | 0.002 |
| DNABERT S | 10k | $0.73 \pm 0.008$ | - | $0.73 \pm 0.011$ | - | - |
| NTv2 50M | 12k | $0.72 \pm 0.005$ | 0.51 | $0.75 \pm 0.008$ | 0.53 | 0.002 |
| NTv2 100M | 12k | $0.73 \pm 0.003$ | 0.51 | $0.76 \pm 0.009$ | 0.56 | 0.002 |
| NTv2 250M | 12k | $0.72 \pm 0.003$ | 0.51 | $0.78 \pm 0.013$ | 0.58 | 0.002 |
| NTv2 500M | 12k | $0.72 \pm 0.003$ | 0.51 | $0.78 \pm 0.009$ | 0.68 | 0.003 |
| HyenaDNA 1K | 1k | $0.71 \pm 0.005$ | 0.51 | $0.63 \pm 0.027$ | 0.49 | 0.002 |
| HyenaDNA 16K | 16k | $0.71 \pm 0.005$ | 0.51 | $0.66 \pm 0.016$ | 0.49 | 0.002 |
| HyenaDNA 32K | 32k | $0.72 \pm 0.002$ | 0.51 | $0.66 \pm 0.012$ | 0.50 | 0.002 |
| HyenaDNA 160K | 160k | $0.71 \pm 0.010$ | 0.51 | $0.56 \pm 0.073$ | 0.49 | 0.002 |
| Caduceus 131K** | 131k | 0.68 | 0.49 | 0.72 | 0.53 | 0.001 |
| Extended NTv2 50M 24K | 24k | $0.72 \pm 0.004$ | 0.51 | $0.75 \pm 0.009$ | 0.53 | 0.002 |
| Extended NTv2 50M 48K | 48k | $0.73 \pm 0.008$ | 0.51 | $0.65 \pm 0.059$ | 0.52 | 0.002 |
| Extended NTv2 50M 96K | 96k | $0.73 \pm 0.006$ | 0.51 | $0.74 \pm 0.019$ | 0.51 | 0.002 |
| Extended NTv2 500M 96K* | 96k | $0.74 \pm 0.004$ | 0.51 | $0.75 \pm 0.018$ | 0.53 | 0.002 |
| GPN-MSA | 128 | $0.70 \pm 0.008$ | 0.55 | $0.97 \pm 0.01$ | 0.97 | 0.35 |
| CNN | 2k | $0.71 \pm 0.003$ | - | $0.58 \pm 0.020$ | - | - |
| Caduceus (from scratch) | 2k | $0.71 \pm 0.003$ | - | $0.61 \pm 0.017$ | - | - |
| Baseline | | $0.76 \pm 0.002$ (Enformer) | 0.56 (CADD) | $0.65 \pm 0.031$ (Enformer) | 0.97 (CADD) | 0.205 (CADD) |

modeling objective, we cannot obtain a probability distriubtion over the tokens that we require to compute zero-shot performance.

Additionally, GPN-MSA was not evaluated on the Bulk RNA, Histone Marks, and DNA Accessiblity tasks. These tasks were processed using the hg19 human genome assembly, and GPN-MSA uses a MSA that aligns to the hg38 human genome assembly. The differences in sequence location and composition between hg19 and hg38 means that applying GPN-MSA to these hg19 requires additional considerations and likely training to do properly. We therefore omit evaluation on these tasks.

### E.2 STATISTICAL SIGNIFICANCE TESTING

To more rigorously asses the perceived gains for each task, we conduct hypothesis testing across all the tasks and models to identify statistically significant differences. For every task, we conduct a Welch's t-Test between each model and every other model. We control for the FDR for the multiple tests by applying Benjamini-Hochberg correction. In Tables 13 - 20 we present the p-values for the models evaluated in the main text on all tasks, rounded to the nearest $1e^{-3}$. For example, this allows the identification of the Enformer baseline having significantly better performance than any of the DNA LMs.

### E.3 COMPUTATIONAL EFFICIENCY

In Table 21, we show the number of parameters for each model, and the FLOPs on an A100 80GB device with batch size 1 for each task category.

### E.4 ADDITIONAL FINE-TUNING ABLATION

In Table 22, we display results for the the full NTv2 series and additional HyenaDNA models. We find that the same pattern discussed in Section 4.4 holds for this larger set of models as well. Namely,

Table 12: Benchmarking performance of DNA LMs and baselines on gene expression prediction, regulatory element, and chromatin features prediction tasks. Models were evaluated in only a fine-tuned setting for this set of tasks. DNABERT-2 was not fine-tuned on the CAGE task due to the incompatibility of the byte pair tokenization with binned labels. GPN-MSA was not fine-tuned on the Bulk RNA, Histone Marks, or DNA Accessibility tasks due to incompatiblities between the genome assembly versions used for the MSA. ** Caduceus 131k results are omitted from the main table due to being limited to an input length of 4k due to computational constraints.

| | Context (bp) | Bulk RNA ($R^2$) | CAGE ($R^2$) | Promoter (AUPRC) | Enhancer (AUROC) | Histone Marks (AUPRC) | DNA Accessibility (AUPRC) |
|---|---|---|---|---|---|---|---|
| | | *Fine-tune* | *Fine-tune* | *Fine-tune* | *Fine-tune* | *Fine-tune* | *Fine-tune* |
| DNABERT-1 | 512 | 0.47 ± 0.007 | 0.14 ± 0.025 | 0.72 ± 0.009 | 0.80 ± 0.005 | 0.23 ± 0.003 | 0.18 ± 0.006 |
| DNABERT-2 | 10k | 0.51 ± 0.050 | - | 0.71 ± 0.112 | 0.81 ± 0.022 | 0.24 ± 0.091 | 0.15 ± 0.064 |
| DNABERT-S | 10k | 0.52 ± 0.060 | - | 0.75 ± 0.021 | 0.83 ± 0.005 | 0.33 ± 0.006 | 0.16 ± 0.039 |
| NTv2 50M | 12k | 0.52 ± 0.074 | 0.35 ± 0.030 | 0.75 ± 0.008 | 0.78 ± 0.041 | 0.34 ± 0.007 | 0.18 ± 0.005 |
| NTv2 100M | 12k | 0.52 ± 0.081 | 0.3 ± 0.030 | 0.78 ± 0.008 | 0.82 ± 0.010 | 0.34 ± 0.007 | 0.22 ± 0.012 |
| NTv2 250M | 12k | 0.57 ± 0.024 | 0.37 ± 0.008 | 0.8 ± 0.008 | 0.84 ± 0.002 | 0.37 ± 0.013 | 0.28 ± 0.006 |
| NTv2 500M | 12k | 0.60 ± 0.038 | 0.39 ± 0.011 | 0.79 ± 0.006 | 0.82 ± 0.002 | 0.38 ± 0.003 | 0.3 ± 0.007 |
| HyenaDNA 1K | 1k | 0.44 ± 0.014 | 0.11 ± 0.015 | 0.7 ± 0.006 | 0.80 ± 0.002 | 0.21 ± 0.001 | 0.13 ± 0.003 |
| HyenaDNA 16K | 16k | 0.46 ± 0.008 | 0.17 ± 0.014 | 0.64 ± 0.004 | 0.75 ± 0.002 | 0.22 ± 0.002 | 0.091 ± 0.003 |
| HyenaDNA 32K | 32k | 0.41 ± 0.012 | 0.22 ± 0.007 | 0.56 ± 0.008 | 0.73 ± 0.001 | 0.22 ± 0.003 | 0.084 ± 0.001 |
| HyenaDNA 160K | 160k | 0.46 ± 0.006 | 0.19 ± 0.032 | 0.67 ± 0.009 | 0.74 ± 0.009 | 0.25 ± 0.004 | 0.11 ± 0.002 |
| Caduceus 131K** | 131k | 0.53 ± 0.008 | 0.163 ± 0.006 | 0.76 ± 0.009 | 0.81 ± 0.006 | 0.10 ± 0.009 | 0.07 ± 0.006 |
| Extended NTv2 50M 24K | 24k | 0.53 ± 0.063 | 0.37 ± 0.010 | 0.75 ± 0.007 | 0.83 ± 0.002 | 0.35 ± 0.007 | 0.19 ± 0.006 |
| ExtendedNTv2 50M 48K | 48k | 0.54 ± 0.038 | 0.36 ± 0.012 | 0.76 ± 0.008 | 0.82 ± 0.002 | 0.35 ± 0.007 | 0.19 ± 0.006 |
| Extended NTv2 50M 96K | 96k | 0.54 ± 0.034 | 0.3 ± 0.019 | 0.76 ± 0.015 | 0.83 ± 0.001 | 0.35 ± 0.005 | 0.19 ± 0.007 |
| Extended NTv2 500M 96K | 96k | 0.56 ± 0.037 | 0.36 ± 0.011 | 0.78 ± 0.003 | 0.82 ± 0.005 | 0.38 ± 0.004 | 0.3 ± 0.006 |
| GPN-MSA | 128 | - | 0.09 ± 0.012 | 0.73 ± 0.015 | 0.79 ± 0.005 | - | - |
| CNN | 2k | 0.42 ± 0.013 | 0.15 ± 0.008 | 0.70 ± 0.002 | 0.78 ± 0.004 | 0.09 ± 0.007 | 0.07 ± 0.003 |
| Caduceus *(from scratch)* | 2k | 0.48 ± 0.048 | 0.10 ± 0.026 | 0.71 ± 0.002 | 0.82 ± 0.012 | 0.09 ± 0.012 | 0.07 ± 0.003 |
| Baseline | | 0.83 ± 0.005 (Enformer) | 0.49 ± 0.000 (Enformer) | 0.86 ± 0.006 (Enformer) | 0.92 ± 0.002 (Enformer) | 0.35 (DeepSea) | 0.44 (DeepSea) |

Table 13: Welch's t-test p-values for causual eQTL task with Benjamini-Hochberg FDR correction.

| | DNABERT-2 | DNABERT-S | NTv2 | NTv2-ext | HyenaDNA-160k | GPN-MSA |
|---|---|---|---|---|---|---|
| DNABERT-S | 0.095 | - | - | - | - | - |
| NTv2 | 1.000 | **0.043** | - | - | - | - |
| NTv2-ext | **0.002** | **0.048** | **0.000** | - | - | - |
| HyenaDNA-160k | 0.125 | **0.012** | 0.080 | **0.0** | - | - |
| GPN-MSA | **0.006** | **0.000** | **0.002** | **0.0** | 0.125 | - |
| Enformer | **0.000** | **0.000** | **0.000** | **0.0** | **0.000** | **0.0** |

full fine-tuning almost uniformly improves model performance relative to partial fine-tuning, by margins that can range up to > 100%. Tasks on which DNA LMs already perform competitively, e.g., regulatory element annotation, seem to benefit less from full-fine tuning, but even here we do see gains. In Table 23 we perform a sensitivity analysis analyze the robustness of our fine-tuning setup to multiple hyperparameter settings, namely for learning rate and batch size. Our analysis shows that

Table 14: Welch's t-test p-values for pathogenic Clinvar task with Benjamini-Hochberg FDR correction.

| | DNABERT-2 | DNABERT-S | NTv2 | NTv2-ext | HyenaDNA-160k | GPN-MSA |
|---|---|---|---|---|---|---|
| DNABERT-S | 0.237 | - | - | - | - | - |
| NTv2 | **0.000** | **0.000** | - | - | - | - |
| NTv2-ext | 0.343 | 0.074 | **0.012** | - | - | - |
| HyenaDNA-160k | **0.001** | **0.001** | **0.000** | **0.0** | - | - |
| GPN-MSA | **0.000** | **0.000** | **0.002** | **0.0** | **0.000** | - |
| Enformer | **0.000** | **0.001** | **0.000** | **0.0** | **0.041** | **0.0** |

Table 15: Welch's t-test p-values for pathogenic Bulk RNA task with Benjamini-Hochberg FDR correction.

|  | DNABERT-2 | DNABERT-S | NTv2 | NTv2-ext | HyenaDNA-160k |
|---|---|---|---|---|---|
| DNABERT-S | 0.782 | - | - | - | - |
| NTv2 | **0.024** | **0.060** | - | - | - |
| NTv2-ext | 0.138 | 0.257 | **0.015** | - | - |
| HyenaDNA-160k | 0.077 | 0.077 | 0.000 | **0.0** | - |
| Enformer | **0.000** | **0.000** | **0.000** | **0.0** | **0.000** |

Table 16: Welch's t-test p-values for the CAGE task with Benjamini-Hochberg FDR correction.

|  | NTv2 | NTv2-ext | HyenaDNA-160k | GPN-MSA |
|---|---|---|---|---|
| NTv2 | - | - | - | - |
| NTv2-ext | **0.003** | - | - | - |
| HyenaDNA-160k | **0.000** | **0.000** | - | - |
| GPN-MSA | **0.000** | **0.000** | **0.000** | - |
| Enformer | **0.000** | **0.000** | **0.000** | **0.000** |

Table 17: Welch's t-test p-values for Promoter task with Benjamini-Hochberg FDR correction.

|  | DNABERT-2 | DNABERT-S | NTv2 | NTv2-ext | HyenaDNA-160k | GPN-MSA |
|---|---|---|---|---|---|---|
| DNABERT-S | 0.479 | - | - | - | - | - |
| NTv2 | 0.184 | **0.005** | - | - | - | - |
| NTv2-ext | 0.233 | **0.020** | **0.016** | - | - | - |
| HyenaDNA-160k | 0.479 | **0.000** | **0.000** | **0.000** | - | - |
| GPN-MSA | 0.703 | 0.159 | **0.000** | **0.000** | **0.000** | - |
| Enformer | **0.024** | **0.000** | **0.000** | **0.000** | **0.000** | **0.000** |

Table 18: Welch's t-test p-values for Enhancer task with Benjamini-Hochberg FDR correction.

|  | DNABERT-2 | DNABERT-S | NTv2 | NTv2-ext | HyenaDNA-160k | GPN-MSA |
|---|---|---|---|---|---|---|
| DNABERT-S | 0.968 | - | - | - | - | - |
| NTv2 | 0.969 | **0.005** | - | - | - | - |
| NTv2-ext | 0.968 | **0.020** | 1.000 | - | - | - |
| HyenaDNA-160k | 0.652 | **0.000** | **0.000** | **0.000** | - | - |
| GPN-MSA | 0.968 | **0.000** | **0.000** | **0.000** | **0.000** | - |
| Enformer | 0.414 | **0.000** | **0.000** | **0.000** | **0.000** | **0.000** |

Table 19: Welch's t-test p-values for the DNA Accessibility task with Benjamini-Hochberg FDR correction.

|  | DNABERT-2 | DNABERT-S | NTv2 | NTv2-ext |
|---|---|---|---|---|
| DNABERT-S | 0.859 | - | - | - |
| NTv2 | **0.002** | **0.000** | - | - |
| NTv2-ext | **0.002** | **0.000** | 1.000 | - |
| HyenaDNA-160k | 0.250 | **0.030** | **0.000** | **0.000** |

Table 20: Welch's t-test p-values for Histone task with Benjamini-Hochberg FDR correction.

|  | DNABERT-2 | DNABERT-S | NTv2 | NTv2-ext |
|---|---|---|---|---|
| DNABERT-S | 0.106 | - | - | - |
| NTv2 | **0.019** | **0.000** | - | - |
| NTv2-ext | **0.019** | **0.000** | 1.000 | - |
| HyenaDNA-160k | 1.000 | **0.000** | **0.000** | **0.000** |

Table 21: Model sizes and FLOPs used per task type.

| Tasks | Hyena 1k (0.6M) | Hyena 16k (1.6M) | Hyena 32k (3.9M) | Hyena 160k (12.9M) | DNABERT-1 (88.6M) | DNABERT-2 (116.6M) | NTv2 50M (50M) | NTv2 100M (100M) | NTv2 250M (250M) | NTv2 500M (500M) |
|---|---|---|---|---|---|---|---|---|---|---|
| Variant Effect | 0.45B | 4.62B | 38.84B | 420.12B | 2.27B | 31.37B | 18.89B | 34.60B | 38.92B | 93.12B |
| Gene Expression | 0.27B | 2.80B | 23.65B | 256.84B | 1.11B | 17.16B | 9.44B | 17.29B | 19.45B | 46.81B |
| Regulatory Element | 0.27B | 2.80B | 23.65B | 256.84B | 1.10B | 17.16B | 9.44B | 17.29B | 19.45B | 46.80B |
| Chromatin Features | 0.27B | 2.80B | 23.65B | 256.84B | 1.10B | 17.16B | 9.44B | 17.29B | 19.45B | 46.80B |

while most results are quite insensitive to hyperparameter choice (with swings $\pm 0.02$ on the metric of interest), users should avoid combinations of higher learning rates (3e-5) and smaller batch sizes (32).

Table 22: Ablation study examining the difference in performance of DNA LM fine-tuning strategies. Results shown correspond to the percent increase in performance of full fine-tuning with respect to freezing LM weights and only training the MLP head.

| | Causal eQTL (AUCROC) | Pathogenic ClinVar (AUROC) | Bulk RNA ($R^2$) | CAGE ($R^2$) | Promoter (AUPRC) | Enhancer (AUROC) | Histone Marks (AUCPRC) | DNA Accessibility (AUPRC) |
|---|---|---|---|---|---|---|---|---|
| NTv2 50M | +1.13 | +9.30 | +30.23 | +71.60 | +1.93 | -2.05 | +32.03 | +33.43 |
| NTv2 100M | +0.98 | +6.24 | +13.70 | +27.72 | +2.16 | +2.83 | +32.70 | +40.54 |
| NTv2 250M | +0.36 | +3.57 | +21.70 | +40.41 | +2.07 | +3.71 | +31.01 | +54.44 |
| NTv2 500M | +0.49 | +4.27 | +24.45 | +42.14 | -1.45 | +0.90 | +22.46 | +47.96 |
| HyenaDNA 1K | +0.95 | +15.39 | +16.50 | +45.22 | +7.13 | +4.68 | +23.61 | +22.65 |
| HyenaDNA 16K | +0.21 | +22.81 | +75.53 | +133.52 | +6.19 | -1.10 | +42.83 | -9.62 |
| HyenaDNA 32K | +0.35 | +11.58 | +82.46 | +102.91 | -18.21 | -6.02 | +14.43 | -22.67 |

Table 23: Fine-Tuning sensitivity analysis on LR and Batch size for Causal eQTL and Bulk RNA tasks.

| Model | LR | Batch size | Causal eQTL (AUCROC) | Bulk RNA ($R^2$) |
|---|---|---|---|---|
| NTv2 500M | $1e^{-5}$ | 32 | $0.723 \pm 0.006$ | $0.597 \pm 0.050$ |
| NTv2 500M | $1e^{-5}$ | 64 | $0.722 \pm 0.003$ | $0.588 \pm 0.048$ |
| NTv2 500M | $1e^{-5}$ | 128 | $0.718 \pm 0.010$ | $0.596 \pm 0.015$ |
| NTv2 500M | $3e^{-5}$ | 32 | $0.717 \pm 0.006$ | $0.580 \pm 0.079$ |
| NTv2 500M | $3e^{-5}$ | 64 | $0.717 \pm 0.007$ | $0.566 \pm 0.016$ |
| NTv2 500M | $3e^{-5}$ | 128 | $0.721 \pm 0.006$ | $0.585 \pm 0.047$ |
| DNABERT 2 | $1e^{-5}$ | 32 | $0.726 \pm 0.005$ | $0.483 \pm 0.135$ |
| DNABERT 2 | $1e^{-5}$ | 64 | $0.719 \pm 0.008$ | $0.503 \pm 0.068$ |
| DNABERT 2 | $1e^{-5}$ | 128 | $0.725 \pm 0.002$ | $0.484 \pm 0.085$ |
| DNABERT 2 | $3e^{-5}$ | 32 | $0.687 \pm 0.067$ | $0.480 \pm 0.063$ |
| DNABERT 2 | $3e^{-5}$ | 64 | $0.713 \pm 0.016$ | $0.507 \pm 0.050$ |
| DNABERT 2 | $3e^{-5}$ | 128 | $0.720 \pm 0.005$ | $0.501 \pm 0.055$ |
| Hyena DNA 160K | $1e^{-5}$ | 32 | $0.703 \pm 0.016$ | $0.459 \pm 0.010$ |
| Hyena DNA 160K | $1e^{-5}$ | 64 | $0.708 \pm 0.010$ | $0.450 \pm 0.006$ |
| Hyena DNA 160K | $1e^{-5}$ | 128 | $0.708 \pm 0.012$ | $0.439 \pm 0.016$ |
| Hyena DNA 160K | $3e^{-5}$ | 32 | $0.701 \pm 0.006$ | $0.456 \pm 0.018$ |
| Hyena DNA 160K | $3e^{-5}$ | 64 | $0.699 \pm 0.010$ | $0.457 \pm 0.006$ |
| Hyena DNA 160K | $3e^{-5}$ | 128 | $0.696 \pm 0.011$ | $0.445 \pm 0.020$ |

## E.5 ADDITIONAL RESULTS BY GENOMIC ANNOTATIONS

In Figure 5, we display additional results from splitting the tasks by genomic annotations.

**Enhancer Detection** We find that DNA LMs have increased performance at identifying enhancers in some repetitive elements, such as LINE1 transposons, as shown in Figure 5a. LINE1 elements are

commonly interspersed along the human genome, and individual LINE1 elements may have uncertain regulatory effects, but DNA LMs appear to be able to call enhancers in LINE1 elements better than in non-LINE1 regions. However, their performance still lags that of the Enformer baseline.

**Zero-shot Pathogenic-ClinVar** In Figure 5b, we observe that most models exhibit increased performance within splice site acceptor regions, with the exception of Enformer, although Enformer demonstrates high performance in both splits.

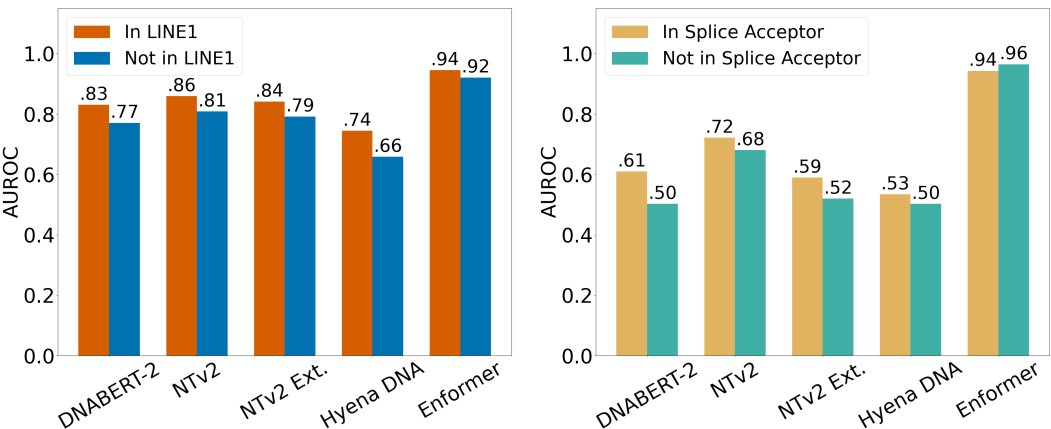

(a) Enhancer detection; split by enhancers located within a LINE1 (transposon) annotation.

(b) Zero-shot Pathogenic ClinVar prediction; by splice site acceptor annotation.

Figure 5: Additional results split by genomic annotations.

### E.6 ADDITIONAL RESULTS FOR EXPERIMENTALLY VERIFIED GENE-PAIRINGS

While the annotations for candidate cis-regulatory elements from SCREEN are reasonable, they nevertheless may include sequences that do not exhibit regulatory activity or ones that only exhibit regulatory activity in specific tissues. As a sanity check, we utilize experimentally verified gene-pairing data for enhancers data from 3D-Chromatin datasets from SCREEN v4. These datasets show allow us to subset our dataset to only use positive labels where we have some sort of experimental signal. Results on this subset are available in Table 24. When using only this experimentally verified subset, the rankings and relative performance of models remain largely unchanged, suggesting that the current form of this dataset is likely still useful for evaluating the relative performance of models at calling regulatory elements.

|  | Original Values (Table 4) | 3D chromatin Verified Enhancers |
|---|---|---|
| DNABERT 2 | $0.80 \pm 0.022$ | $0.83 \pm 0.002$ |
| NTv2 500M | $0.82 \pm 0.002$ | $0.86 \pm 0.002$ |
| HyenaDNA 160k | $0.74 \pm 0.009$ | $0.77 \pm 0.002$ |
| Enformer | $0.92 \pm 0.001$ | $0.95 \pm 0.001$ |

Table 24: Enhancer task update results.

## F ASSETS

In Table 25, we list the open source libraries and repositories used in this work, with their corresponding licenses.

Table 25: Open source libraries (and corresponding licenses) used in this work.

| Library | License |
| --- | --- |
| Biopython (Cock et al., 2009) | Biopython license |
| Haiku (Hennigan et al., 2020) | Apache 2.0 |
| HuggingFace (Wolf et al., 2019) | Apache 2.0 |
| Jax (Bradbury et al., 2018) | Apache 2.0 |
| Jupyter (Kluyver et al., 2016) | BSD 3-Clause |
| NumPy (Harris et al., 2020) | NumPy license |
| Matplotlib (Hunter, 2007) | Matplotlib license |
| Pandas (The pandas development team, 2020) | BSD 3-Clause "New" or "Revised" |
| Optax  (DeepMind et al., 2020) | Apache 2.0 |
| PyFaidx (Shirley et al., 2015) | BSD-3-Clause |
| PyTorch (Paszke et al., 2019) | BSD-3 Clause |
| Scikit-Learn (Pedregosa et al., 2011) | BSD 3-Clause |
| Seaborn (Waskom, 2021) | BSD 3-Clause "New" or "Revised" |
| TensorFlow (Abadi et al., 2015) | Apache 2.0 |

## G  COMPUTATIONAL RESOURCES

All research in this study was conducted using Cloud TPU's provided by Google's TPU Research Cloud program. Specifically, a TPU-v4-64 slice was used for all context length extension pre-training. Single TPU-v4 machines were used in parallel to conduct all benchmarking and evaluations including fine-tuning, zero-shot, and inference experiments.

## IMPACT STATEMENT

As our work introduces a benchmark, we do not believe it poses any inherent negative societal impacts. In fact, our work will hopefully create a positive impact by accelerating the development of useful DNA LMs that can bring about a deeper understanding of biology.

