# OpenReview forum: "The Human Genomics Long-Range Benchmark: Advancing DNA Language Models"
_ICLR.cc/2026/Conference — Submitted to ICLR 2026_

### Official Review · Reviewer_oywS · 2025-10-30

**Soundness:** 3
**Presentation:** 4
**Contribution:** 2
**Rating:** 6
**Confidence:** 4

**Summary:**

The authors presented a new benchmark for evaluating DNA language models focused on the study of context sizes. The authors compiled a diverse set of long-range and short-range downstream tasks for DNA LMs such as variant effect prediction, gene expression, regulatory element and chromatin feature predictions. The tasks and datasets are clearly described. And it comes with some user-friendly features such as variable context size versions and visualization tools. Overall, I find it to be one of the better DNA LM benchmarks with a broad range of tasks and specific aims. I therefore recommend acceptance.

**Strengths:**

1. The authors selected a diverse and biologically meaningful array of downstream tasks.
2. The benchmark focuses on the study of context sizes, which is an important topic in DNA LM development and application. This benchmark can be expected to provide valuable insights to the field.
3. Details of the datasets are thoroughly documented.
4. The visualization tool looks helpful.

**Weaknesses:**

1. DART-Eval is another good DNA LM benchmark, but not mentioned in the paper. The authors should discuss how they are different and in which aspects LRB is potentially more useful.
2. More results on Evo-2 would be beneficial, since it has shown significant improvement in performance compared with previous DNA LMs. Would it be feasible for the authors to evaluate the transfer learning capability of it (i.e. fine-tuning), even with the 1B model?
3. It would be better to highlight the difference between ClinVar and OMIM benchmarks (missense vs. non-coding) in the task names, etc.

**Questions:**

1. In Table 3, Enformer is not a good baseline for the ClinVar task, since this set only contains missense variants, and Enformer is a model focused on the non-coding genome.
2. In Table 4, why is DeepSEA chosen over Enformer for the last two tasks?

---

> ### Author Response · Authors · 2025-11-25
> **Response to Reviewer oywS**
>
> We thank the reviewer for their comments and feedback.
>
>  > W 1: Provide comparison to DART
>
> We thank the reviewer for this suggestion and will add this benchmark to our updated manuscript, as indeed the DART-eval improves upon existing work in terms of biological relevance of tasks and is a very good benchmark for short-range tasks. However our LRB tasks are still more comprehensive in terms of number of tasks, type of tasks and amount of data, and for our long-range tasks we have the unique ability to download contexts of any size. (Using the nomenclature of our Table 1, we would designate DART as having check marks for human-centricity and biological relevance, but an X mark for long-range).
>
>  > W 2: More Evo2 results
>
> Due to computational constraints we were unable to fine-tune any variant of Evo2 with the resources available to us. Evo2 requires FP8 support requiring a Hopper architecture card (such as an h100). While we were able to run zero-shot evals, we do not have enough h100s available to be able to fine-tune even the smallest (1B parameters) version of evo2 at the moment.  We hope to be able to update this work with numbers for Evo2 in the future, but due to Evo2’s computational cost and infrastructure requirements we are unable to include it at this time in any capacity besides zero-shot evaluation.
>
>  > W 3 Highlight differences between ClinVar and OMIM
>
> Thank you for the feedback, we will adjust the naming to better reflect the types of variants captured in each task.
>
> > Q 1 Enformer for ClinVar
>
> We used Enformer as a supervised baseline for ClinVar since it outperformed two other supervised baselines we trained (Table 11 in the appendix), however we note that comparisons on the supervised version of the ClinVar tasks should be directed towards GPN-MSA due to its very high performance on that task. We include Enformer purely for completeness.
> |                    |       CNN     |       Caduceus (from scratch)     |      Enformer      | GPN-MSA |
> |:------------------:|:--------------:|:--------------:|:--------------:|:---------:|
> | eQTL (auROC)       | 0.71 +- 0.03   | 0.71 +- 0.03   | 0.76 +- 0.002  | 0.97 +- 0.01 |
>
>
> > Q 2 DeepSEA vs Enformer
>
> This was an empirical choice, on the dataset we used Enformer performed worse on these tasks.
> We list DeepSEAs performance on these tasks over Enformer again due to DeepSEAs better performance on these tasks.
>
> |                    |       Enformer 1k       |      Enformer 16k      |      Enformer 65k      |     Baseline    |
> |:------------------:|:--------------:|:--------------:|:--------------:|:---------------:|
> | DNA Access (AUPRC) | 0.101 +- 0.01  | 0.09 +- 0.001  | 0.082 +- 0.006 | 0.44 (Deep SEA) |
> | Histone (AUPRC)    | 0.145 +- 0.001 | 0.196 +- 0.002 | 0.282 +- 0.001 | 0.35 (Deep SEA) |

---

### Official Review · Reviewer_gKNB · 2025-11-01

**Soundness:** 3
**Presentation:** 3
**Contribution:** 2
**Rating:** 4
**Confidence:** 4

**Summary:**

The paper presents a benchmark for assessing the quality of genomic language models, focusing on long-range dependencies. The datasets can be extracted from the benchmark at different context lenths, lowering the barier for conducting experiments at different context-length scales. Compared to earlier benchmarks, the paper also investigates more elaborate fine-tuning techniques, demonstrating improvements over earlier protocols. Finally, an analysis and visualization tool is included. Based on the benchmarks, the authors draw some conclusions on the state of the field.

**Strengths:**

**Quality** and **Clarity**. The authors motivate the problem well, making an effort to describe earlier work, and how it differs from the current contribution. The nine prediction tasks are presented at a reasonable level of detail, explicitly stating the biologically relevance and the long-range dependencies of each task.

**Significance** The paper addresses an important problem. The last years have seen a considerable increase in genomic language models, but it still remains unclear how well they perform compared supervised task-specific models. Comparing genomic language models head-to-head is also difficult - hampered by the fact that papers often report results on different experiments. It is therefore important to establish meaningful benchmarks to move the field forward.

**Originality** The paper introduces tasks that are believed to depend on long-range genomic interactions, which have been insufficiently addressed in previous benchmarks.

**Weaknesses:**

The main weakness of the paper is that it is seems fairly incremental. Several benchmarks for genomic language models already exist, and the current paper claims to distinguish itself by being biologically meaningful and focusing on long-range dependencies, but the BEND benchmark made similar claims when it came out. The other difference - partial fine-tuning vs full model fine-tuning - are interesting, but such observations would not normally be considered sufficient for a paper in a top machine learning conference.

As far as I can see, most of the tasks are copied from earlier benchmarks, and simply provided here with longer context lengths. Since this benchmark focuses on long-range effects, one would expect the authors to demonstrate that long range effects are critical in each of the nine sets, but they only seem to do so for the Bulk RNA task (figure 3).

More generally, we are seeing multiple new benchmarks for DNA language models coming out, and it is unclear to me whether they keep providing value to the community. One could argue that it would be more fruitful if a community-wide effort was made to consolidate to a set of agreed-upon tasks, rather than making incremental updates to existing ones. For example, another long-range benchmark called DNALongBench was recently proposed with similar benchmarks (but with some metrics that more directly addressed long-range dependencies).

**Questions:**

### Questions
The Bulk RNA task in Figure 3 shows clear context length dependencies. Do the other eight tasks show similar dependencies? And if not, does this mean that the long range effects stated in the "Long-Range" sections for the individual prediction tasks are not as critical as expected - or that current models cannot pick them up?

Table 4. Why was Evo2 not included here?

line 391. *"For DNA LMs to be useful for these tasks, they must also find a way to model and learn evolutionary pressures and conservation."*. The concluding statement was not quite clear to me. Why can't we just use the GPN-MSA model (or the more recent GPN-Star).

lin 451. *"most DNA LMs saw an increase in performance with increasing context"*. Does the increase in context window also result in more fine-tuned parameters, and if so, could this be a potential reason for the improved performance?

### Minor comments

line 056: the authors state *"Allowing users to select arbitrary sequence length inputs for any given dataset enables us for the first time to understand empirically the importance of long-range inputs for our proposed tasks."*. As far as I remember the BEND dataset also offered the possibility of extracting arbitrary flanking regions, so the "first time" in this sentence should perhaps we rephrased.

line 148. *"GTEx"*. Has this term been introduced?

line 213. Is it intentional that the "Pathogenic Clinvar" section has no "Long-Range" subsection?

line 230. *"are can advance"*. Remove "are".

line 236. *"Outputs are RPKM normalized"*. Has "RPKM" been introduced?

line 318. It would be helpful if the authors could make it clearer why GPN-MSA is considered a baseline, rather than a language models on par with the others. Is the use of MSAs incompatible with particular downstream tasks?

---

> ### Author Response · Authors · 2025-11-25
> **Response to Reviewer gKNB 1/2**
>
> > Weakness 1: Comparison to Previous Work
>
> We position this work as expanding on long-context tasks, since previous work has been lacking on modeling long range interactions at all (e.g. BEND only has two tasks with more than 512bp inputs and only ~100 datapoints for each at test time). Concurrent work like DNALongBench also identifies the need to model long range interactions and additionally propose a contact map prediction task, however their evaluation is severely limited both in terms of models tested (only testing HyenaDNA and Caduceus, two of the weaker models in our evaluation) and in terms of dataset size (only ~400 datapoints in the val/test splits from 5 cell types in the main text, and 4 more cell types in the supplemental), leading to possible issues with variance and technical noise while still requiring a large amount of compute to run. In our work, we explore a much more models (for example, including NTv2, DNABERT, and Evo2 zero-shot results) at varying context lengths more clearly establishing some of the scaling behaviour with context size (Fig 3) and also conduct multiple replications to provide error bars for our analysis.
>
> To summarize, we aimed to present a benchmark that:
> * Captures multiple properties of interest, including long range interactions which have been under studied in previous works
> * Allows users to compare performance on these tasks at multiple context lengths instead of only at fixed lengths
> * Provides reasonable recipes for fine-tuning to enable comparisons
> * Sets reasonable baselines (where most DNA LMs lose to our selected baselines)
> * Includes annotations so we can assess post-hoc what kind of failure modes models encountered.
>
>
>
>
> > Weakness 2: Not all tasks are shown to be long context
>
>
> More than just the bulk RNA task is long range, we find that the CAGE task and the Histone task also exhibit similar signals. To show this, we conduct fine-tuning experiments on eight tasks using Enformer at 3 different context lengths (1k, 16k, 65k bps) to show how each task performs when trained on various context lengths (attached below).
> |   |       1k       |       16k      |       65k      |     Baseline    |
> |:----------:|:--------:|:---------:|:--------:|:------:|
> | eQTL (auROC)       | 0.73 +- 0.01   | 0.72 +- 0.02   | 0.76 +- 0.002  | - |
> | ClinVar (auROC)    | 0.72 +- 0.01   | 0.72 +- 0.01   | 0.74 +- 0.03   | 0.97 (GPN-MSA)  |
> | RNA (R²)           | 0.67 +- 0.004  | 0.802 +- 0.001 | 0.83 +- 0.002  | -    |
> | Cage (R²)          | 0.359 +- 0.004 | 0.456 +- 0.002 | 0.472 +- 0.002 | -   |
> | Prom (AucPR)       | 0.842 +- 0.001 | 0.862 +- 0.021 | 0.866 +- 0.006 | -  |
> | Enhancer (AUROC)   | 0.89 +- 0.001  | 0.901 +- 0.02  | 0.906 +- 0.02  | -  |
> | DNA Access (AUPRC) | 0.101 +- 0.01  | 0.09 +- 0.001  | 0.082 +- 0.006 | 0.44 (Deep SEA) |
> | Histone (AUPRC)    | 0.145 +- 0.001 | 0.196 +- 0.002 | 0.282 +- 0.001 | 0.35 (Deep SEA) |
>
> We see clear long context interactions in the bulk RNA and CAGE prediction tasks, and interestingly we also see a context length effect on the Histone task, a task where the baseline DeepSEA method only sees 1k bps at input but the only tasks where DNA LMs actually outperformed the existing baseline.
>
> However, we want to stress that not all tasks are intended to be long range. We included a mix of tasks focused on capturing a variety of properties.
>
> For example, in tasks  like variant effect prediction in ClinVar, models like GPN-MSA achieve SOTA performance only on 128 input BPs due to including MSA information directly. We include tasks like these to provide useful watermarks to assess how models evolve on multiple axes, for example being able to identify decreasing performance on conservation based tasks like ClinVar as we increase context lengths might be useful to know.
>
> To summarize, we do have more than a single task that capture long range interactions, and furthermore we intentionally include tasks that capture different genomic properties to assess multiple model qualities for practitioners using this for evaluation.

---

> ### Author Response · Authors · 2025-11-25
> **Response to Reviewer gNKB 2/2**
>
> > Question 1: Effect of longer context in the other 8 tasks
>
>
> We discuss the context length dependence in W2 above. To the reviewers point about disambiguating between the importance of long-range signal and the abilities of models to actually pick up these signals when present, we conducted evaluation at various context lengths for the Enformer baseline. We chose Enformer as the baseline for many tasks because of its track record in modeling long range interactions as shown in Figure 3 where we see that the Enformer baseline monotonically increases with context lengths while the other models saturate, this leads us to believe that at least in the BulkRNA task:
>  1. There is signal coming from longer interactions
>  2. existing DNA LMs are unable to adequately match this signal as all of them (even those with more parameters) are strictly worse than the Enformer baseline.
>
> In general, it can be hard to disambiguate between the lack of long range signal and the models inability to capture long range signal, but in this work we believe that the positive correlation of Enformer with context length should be sufficient to claim that there is sufficient long range signal that other models should also be able to capture.
>
> > Q 2: Lack of Evo2 in Table 4
>
> Table 4 is only fine-tuned tasks, and due to computational constraints we were unable to fine-tune any variant of Evo2 with the resources available to us. Evo2 requires FP8 support requiring a Hopper architecture card (such as an h100). While we were able to run zero-shot evals, we do not have enough h100s available to be able to fine-tune even the smallest (1B parameters) version of evo2 at the moment.  We hope to be able to update this work with numbers for Evo2 in the future, but due to Evo2’s computational cost and infrastructure requirements we are unable to include it at this time in any capacity besides zero-shot evaluation.
>
> > Q 3: Statement about GPN-MSA/GPN-STAR
>
> We position this work as a benchmark specifically for evaluating LLMs in this space, and specifically for variant effect prediction tasks GPN-MSA and its successors are a fantastic target that DNA LMs should theoretically be able to match.  Our statement was referencing the fact that DNA LMs should be able to approach GPN-MSAs performance and that therefore GPN-MSA is a concrete target that can be aimed for (and already Evo2 seems to be approaching this). For users aiming to do variant effect prediction in species where GPN-MSA and GPN-STAR work, practitioners should of course use the more effective model until DNA LMs are able to catch up to them. However there are use cases where single sequence DNA LMs might make a more compelling choice over alignment based models, for example in regions or species with no high quality MSA available.
>
> > Q 4: Are Parameter counts different for each context length
>
>
> No, the context window and parameter count are completely independent in our analysis. For example, in Figure 3 every instance of Enformer was 287M parameters, each instance was only trained on a different context window size. The performance increases are therefore a result of the increased context length. Furthermore, we see that models like NT-v2 (50M and 500M) are strictly worse than the enformer baseline at all context lengths, even the one with nearly twice as many parameters.
>
> ### Minor Points
> * Thank you for the minor corrections.
> * GTEx is the name of the project we source some data from.
> * Yes, as stated above and shown by GPN-MSAs performance on the ClinVar task, this task does not rely on long range interactions as much as other tasks, we include it to maintain tasks on other scales of interaction to more holistically evaluate models.
> * RPKM is Read-Per Kilobase-Million and is a normalization strategy for this type of data, while it is a standard term in bioinformatics literature we will clarify.
> * We classify GPN-MSA differently because it has additional inputs at inference time (the alignments from the other species in its alignment). We split it off into its own category (alignment based LM) to clarify the difference between single sequence LMs. GPN-MSA is only able to be evaluated where its MSA has support, for example some tasks are based on sequences from a different assembly we cannot easily query the alignment for those tasks.

---

### Official Review · Reviewer_WR37 · 2025-11-01

**Soundness:** 3
**Presentation:** 3
**Contribution:** 3
**Rating:** 8
**Confidence:** 2

**Summary:**

This paper presents the Human Genomics Long-Range Benchmark (LRB), which includes nine curated tasks to test how well DNA foundation models can handle long-range sequence reasoning. The benchmark combines data from well-known public sources such as ENCODE, FANTOM5, GTEx, and SCREEN, covering tasks like regulatory element detection, chromatin state classification, and gene expression prediction. One special but simple feature is that users can download DNA sequences of any length, because each task is defined by genomic coordinates instead of fixed-size sequence windows. This design allows fair comparison between models with different input sizes (for example, 2 kb vs. 131 kb) and makes it easier to study how model performance changes with longer contexts—something that older benchmarks could not support.

The authors also standardize all data to the GRCh38 reference genome, provide a visualization notebook and fine-tuning scripts, and show that full-parameter fine-tuning often gives better results than parameter-efficient tuning methods. Their experiments find that current DNA language models do well on local annotation tasks but still perform worse than supervised models like Enformer on gene expression prediction and alignment-based models on zero-shot variant effect tasks.

In summary, the paper does not propose a new model but makes an important contribution through a clear, flexible, and easy-to-use benchmark. The LRB transforms a simple data design idea into a useful and reproducible platform for studying long-range reasoning in human genomics.

**Strengths:**

1. Clear problem focus.
The paper clearly explains that most existing DNA language model benchmarks use short sequences and do not capture deeper biological meaning. LRB instead focuses on long-range genomic effects, which are more realistic and biologically important.

2. Well-defined tasks and data. (curated / repurposed from public available datasets)
The benchmark includes a diverse set of human-centered tasks with clear goals and detailed definitions. Each task is designed to test specific biological hypotheses related to long-range regulation.

3. Practical methodological insight.
The study shows that full fine-tuning improves model performance compared to head-only and PEFT, and recommends a clear procedure for others to follow. (however more in-depth discussion of why this is the case will further strengthen the paper more)

4. Helpful tools.
The provided visualization notebook and flexible sequence downloader make it easier to analyze errors, test different context lengths, and better understand model behavior.

5. Transparent evaluation.
The paper clearly reports where current DNA language models succeed and where they fall short: especially on long-range gene expression and zero-shot variant effect tasks, which highlighting the remaining performance gap with models like Enformer and alignment-based approaches. This helps set realistic benchmarks for future work.

**Weaknesses:**

1. Limited hyper-parameter tuning.
The authors pointed out the hyperparameter search is minimal, which can affect the ranking between models. Since the benchmark includes architectures of very different sizes and training dynamics, some performance differences might reflect suboptimal tuning rather than true capability gaps. (But this does not affect the quality of the benchmark itself, so a minor weakness)

2. Compute limitations.
Large models such as Evo2 are only tested in zero-shot mode on a subset of tasks. This limits how confidently we can generalize the results to those DNA models, and makes it difficult to evaluate whether fine-tuning scales consistently with model size. (again, not a weakness on the benchmark, but on the authors provided example results using the benchmark)

3. Task scope bias.
Although LRB is designed for long-range reasoning, most tasks involve bulk measurements or are centered on transcription start sites (TSS). This framing naturally benefits models like Enformer, which were trained with similar positional biases, and may underrepresent more complex regulatory dependencies such as distal enhancer–promoter interactions.

4. Single-species focus.
The benchmark only includes human genomic data, without any cross-species or multi-genome comparisons.

-- note on point 3 & 4 above --
My background is not biomedicine and my knowledge in this domain is limited. I cannot confidently judge how well the selected tasks covers or not covers all the bases.

5. Mixed training paradigms.
The evaluation compares self-supervised DNA LMs (trained with masked modeling) to supervised architectures like Enformer. While informative, this comparison mixes objectives and data regimes. A clearer discussion of how these paradigms differ and whether the gap reflects modeling power or training signal would strengthen the conclusions.

**Questions:**

1. On fine-tuning strategy:
a. The insight on fine-tuning strategy is intriguing, the reasoning given in the paper makes sense but is very brief (may be limited by the page limit). Have you analyzed which layers or representations benefit most from full fine-tuning compared to PEFT? (e.g. start by enabling LoRA  only on the top few transformer blocks, then progressively include deeper layers (e.g., top 2 → top 4 → top 8 → all layers).
b. Could adapter-style methods (e.g., LoRA or prefix tuning) recover similar gains if tuned for longer or with task-specific regularization?

2. On context-length scaling:
a. How do performance trends behave beyond 131 kb? Is there an observed saturation point where longer context no longer improves performance?
b. When users extract longer sequence windows (e.g., extending from 2 kb to 131 kb around a labeled locus), how does the benchmark ensure that these expanded regions do not overlap with labeled sites from the validation or test sets?

3. On task coverage and bias:
Each selected task is clearly described, but many are promoter-centered or TSS-centered, which mainly capture local regulatory signals. How do the authors ensure that these tasks are representative of broader long-range genomic phenomena? Have you considered including tasks such as enhancer–promoter interaction or 3D chromatin contact prediction, which also reflect truly long-range dependencies? My understanding of genomics is limited, so I am unsure whether the current task set fully covers the important biological contexts.

**Details Of Ethics Concerns:**

no concern.

---

> ### Author Response · Authors · 2025-11-25
> **Response to WR37 Part 1**
>
> We thank the reviewer for their comments and feedback.
>
> > W 1. Lack of Hyperparam tuning
>
>
> Unfortunately, given the breadth of models and tasks, we were not able to perform a full hyperparameter sweep for the main results in our work. However, in the appendix Table 23, (reproduced below) we provide a sensitivity analysis for learning rate and batch size across three model classes (NTv2, DNABERT2, HyenaDNA) and two of the tasks in our benchmark (causal eQTL variant effect prediction and BulkRNA prediction). Our analysis shows that most results are quite insensitive to hyperparameter choice (with swings $\pm$ 0.02 on the metric of interest).
> | Model          | LR        | Batch size | Causal eQTL   | Bulk RNA    |
> |----------------|-----------|------------|---------------|---------------|
> | NTv2 500M      | $1e^{-5}$ | 32         | 0.723 ± 0.006 | 0.597 ± 0.050 |
> | NTv2 500M      | $1e^{-5}$ | 64         | 0.722 ± 0.003 | 0.588 ± 0.048 |
> | NTv2 500M      | $1e^{-5}$ | 128        | 0.718 ± 0.010 | 0.596 ± 0.015 |
> | NTv2 500M      | $3e^{-5}$ | 32         | 0.717 ± 0.006 | 0.580 ± 0.079 |
> | NTv2 500M      | $3e^{-5}$ | 64         | 0.717 ± 0.007 | 0.566 ± 0.016 |
> | NTv2 500M      | $3e^{-5}$ | 128        | 0.721 ± 0.006 | 0.585 ± 0.047 |
> | DNABERT 2      | $1e^{-5}$ | 32         | 0.726 ± 0.005 | 0.483 ± 0.135 |
> | DNABERT 2      | $1e^{-5}$ | 64         | 0.719 ± 0.008 | 0.503 ± 0.068 |
> | DNABERT 2      | $1e^{-5}$ | 128        | 0.725 ± 0.002 | 0.484 ± 0.085 |
> | DNABERT 2      | $3e^{-5}$ | 32         | 0.687 ± 0.067 | 0.480 ± 0.063 |
> | DNABERT 2      | $3e^{-5}$ | 64         | 0.713 ± 0.016 | 0.507 ± 0.050 |
> | DNABERT 2      | $3e^{-5}$ | 128        | 0.720 ± 0.005 | 0.501 ± 0.055 |
> | Hyena DNA 160K | $1e^{-5}$ | 32         | 0.703 ± 0.016 | 0.459 ± 0.010 |
> | Hyena DNA 160K | $1e^{-5}$ | 64         | 0.708 ± 0.010 | 0.450 ± 0.006 |
> | Hyena DNA 160K | $1e^{-5}$ | 128        | 0.708 ± 0.012 | 0.439 ± 0.016 |
> | Hyena DNA 160K | $3e^{-5}$ | 32         | 0.701 ± 0.006 | 0.456 ± 0.018 |
> | Hyena DNA 160K | $3e^{-5}$ | 64         | 0.699 ± 0.010 | 0.457 ± 0.006 |
> | Hyena DNA 160K | $3e^{-5}$ | 128        | 0.696 ± 0.011 | 0.445 ± 0.020 |
>
>
> > W 2. Compute Limitations: Evo2 tested only zero-shot
>
>
> We thank the reviewer for understanding these limitations.
> We hope to be able to update this work with numbers for Evo2 in the future, but due to Evo2’s computational cost and infrastructure requirements we are unable to include it at this time in any capacity besides zero-shot evaluation.
>
> > W 3. Task scope bias
>
>
> This is unfortunately a limitation of the field: there are many different ways that long range interactions can manifest in the genome, and we cover a subset (e.g., interactions mediated in cis likely governed by sequence motifs and variants). While there have been recent advances in Chromatin Conformation Capture experiments that may be of interest here, these datasets are generally small, as they are generated in individual cell types and their resolution is such that at the smallest end the inputs are 1MB and the outputs are interactions between 2k chunks of the genome. For example, in concurrent work like DNALongBench also identifies the need to model long range interactions and additionally propose a contact map prediction task, however their evaluation is severely limited both in terms of models tested (only testing HyenaDNA and Caduceus, two of the weaker models in our evaluation) and in terms of dataset size (only ~400 datapoints in the val/test splits from 5 cell types in the main text, and 4 more cell types in the supplemental), leading to possible issues with variance and technical noise while still requiring a large amount of compute to run. While in the future we can see additional data like this being useful,  In our work, we chose to focus on exploring a much larger range of models at varying context lengths more clearly establishing some of the scaling behaviour with context size (Fig 3) and also conducting multiple replication to provide error bars for our analysis.
>
> > W 4. Only Human Data
>
>
> We specifically focus on human data due to the relatively large amount of high quality data and annotations in humans compared to other species.
>
> Moreover, several of the prominent DNA language models (e.g, NT, HyenaDNA, Caduceus) were trained on only human genome data. We expect this human-focused trend to continue in the community and therefore see our benchmark as a useful test-bed for existing and future models.

---

> > ### Author Response · Authors · 2025-11-25
> > **Response to WR37 Part 2**
> >
> > > W 5. Mixed Paradigm
> >
> >
> > We include supervised methods like Enformer as a baseline, since it’s a generally accepted model in this space with strong performance on many tasks. Given its supervised training on the similar tasks, Enformer can be seen as a sort of ‘upper bound’ watermark on the performance of DNA LMs that were pre-trained in a self-supervised manner. Even though Enformer is not pre-trained like DNA LMs,  we fine-tune all models (all DNA LMs and Enformer) in the same fashion on the same data and observe that Enformer outperforms all tested DNA LMs on a majority of tasks. We therefore believe that Enformer is a reasonable target, given the same input context DNA LMs should be able to reach a similar level of performance to Enformer.
> >
> >
> >
> > > Q 1.Fine Tuning
> >
> >
> > We did not conduct the probing as described here. While we might expect to uncover something similar to the the reviewer’s hypothesized findings (e.g. certain layers exhibiting the most concrete gains from full FT vs PEFT), out work is mainly focused on enabling direct comparisons between model performance (and where they fail) and not on identifying sub-components of models in this manner. Users can certainly use our benchmark and datasets however to conduct this analysis.
> > Even when doing LoRA fine-tuning, we saw plateauing performance on the validation step, which we then early-stopped, leading us to conclude that continuing training would likely not result in large performance differences. Task specific regularization might help in achieving higher absolute performance, but for our purposes we chose to focus on ensuring fair comparison between evaluated models.
> >
> > > Q 2 Context length scaling
> >
> >
> > We extended Enformer past 190kb on the bulk RNA task, and we found that performance started saturating at ~130 kb. This could be due to the model not being able to pick up signal past that or that there was little information gain.
> > Our datasets are always split by chromosome, so expanding context in the training will never overlap and start returning full regions that are in the val or test dataset. Additionally, while there may still be some homologous between subsequences labels are still never exposed, but even then we additionally conduct homology based splitting like Enformer to further try to avoid this.
> >
> > > Q 3 Task bias
> >
> >
> > See above in W3

---

### Author Response · Authors · 2025-11-25
**General Response**

We’d like to thank all the reviewers for their comments and feedback.
We addressed each reviewer's concerns in individual comments, here we document some of the major additions we have made.

>Addition 1: More Results by context length

We add in a new table showing performance by different context lengths on all tasks, showing that tasks like bulk RNA, CAGE, and Histones clearly scale with context length. While this is only with a single model at the moment (Enformer), we believe this clearly shows how multiple tasks clearly scale with context length, while others (such as ClinVar) remain relatively constant across context lengths, likely due to most of their signal coming from conservation. We reiterate that this is by design, as we aimed to include tasks from multiple context scales to more holistically evaluate DNA LMs

|                    |       1k       |       16k      |       65k      |     Baseline    |
|:------------------:|:--------------:|:--------------:|:--------------:|:---------------:|
| eQTL (auROC)       | 0.73 +- 0.01   | 0.72 +- 0.02   | 0.76 +- 0.002  | -               |
| ClinVar (auROC)    | 0.72 +- 0.01   | 0.72 +- 0.01   | 0.74 +- 0.03   | 0.97 (GPN-MSA)  |
| RNA (R²)           | 0.67 +- 0.004  | 0.802 +- 0.001 | 0.83 +- 0.002  | -               |
| Cage (R²)          | 0.359 +- 0.004 | 0.456 +- 0.002 | 0.472 +- 0.002 | -               |
| Prom (AucPR)       | 0.842 +- 0.001 | 0.862 +- 0.021 | 0.866 +- 0.006 | -               |
| Enhancer (AUROC)   | 0.89 +- 0.001  | 0.901 +- 0.02  | 0.906 +- 0.02  | -               |
| DNA Access (AUPRC) | 0.101 +- 0.01  | 0.09 +- 0.001  | 0.082 +- 0.006 | 0.44 (Deep SEA) |
| Histone (AUPRC)    | 0.145 +- 0.001 | 0.196 +- 0.002 | 0.282 +- 0.001 | 0.35 (Deep SEA) |
> Addition 2: Updated performance metrics

During the rebuttal period, we were able to update some of our metrics, for example Evo7B on the clinvar task 0.89 -> 0.92 and Enformer on ClinVar 0.65 -> 0.74.

While multiple reviewers asked about adding fine-tuned versions of Evo2, we currently do not anticipate being able to include those during this rebuttal period due to computational constraints. However, we believe we have clearly established an evaluation methodology here which allows us to evaluate Evo2 at a later date when we or another member of the field is able to allocate sufficient computational resources to this.

---

### Meta-Review · Area_Chair_hHFn · 2026-01-06

**Summary:**

The primary concerns informing the borderline decision center on the study's perceived incrementality and critical experimental limitations. Reviewer gKNB argued that the benchmark lacks sufficient differentiation from existing suites like BEND and DNALongBench, questioning the novelty of the contribution given that clear long-range dependency benefits were only empirically demonstrated for the Bulk RNA task. Multiple reviewers (oywS and WR37) flagged significant evaluation gaps, including the absence of comparisons to DART-Eval and the omission of fine-tuning results for state-of-the-art models like Evo2 due to computational constraints, which limits the benchmark's ability to definitively assess the capabilities of the most advanced long-context architectures.

**Reviewer Concerns:**

The most significant outstanding concern remains the omission of fine-tuning results for Evo2; the authors cited computational constraints for excluding this state-of-the-art model, leaving a critical gap in verifying whether the most advanced architectures can actually leverage the benchmark's long-context design.

**Reviewer Scores:**

Reviewer WR37 would likely have maintained their high score of 8, as the authors' sensitivity analysis effectively addressed their minor concerns regarding hyperparameter tuning. Reviewer gKNB (score 4) and Reviewer oywS (score 6) would have been the most likely to maintain the score. As noted by both gKNB and oywS, the paper is not yet ready for publication because, despite claiming to be a long-range benchmark, it ignores the most important baseline Evo2.

---

### Decision · Program_Chairs · 2026-01-26

Reject